# *VL-SAE*: Interpreting and Enhancing Vision-Language Alignment with a Unified Concept Set

**Shufan Shen**[1,2]   **Junshu Sun**[1,2]   **Qingming Huang**[1,2]   **Shuhui Wang**[1]*

[1]State Key Lab. of AI Safety, Institute of Computing Technology, Chinese Academy of Sciences
[2]University of Chinese Academy of Sciences
{shenshufan22z, sunjunshu21s, wangshuhui}@ict.ac.cn   qmhuang@ucas.ac.cn

## Abstract

The alignment of vision-language representations endows current Vision-Language Models (VLMs) with strong multi-modal reasoning capabilities. However, the interpretability of the alignment component remains uninvestigated due to the difficulty in mapping the semantics of multi-modal representations into a unified concept set. To address this problem, we propose VL-SAE, a sparse autoencoder that encodes vision-language representations into its hidden activations. Each neuron in its hidden layer correlates to a concept represented by semantically similar images and texts, thereby interpreting these representations with a unified concept set. To establish the neuron-concept correlation, we encourage semantically similar representations to exhibit consistent neuron activations during self-supervised training. First, to measure the semantic similarity of multi-modal representations, we perform their alignment in an explicit form based on cosine similarity. Second, we construct the VL-SAE with a distance-based encoder and two modality-specific decoders to ensure the activation consistency of semantically similar representations. Experiments across multiple VLMs (*e.g.*, CLIP, LLaVA) demonstrate the superior capability of VL-SAE in interpreting and enhancing the vision-language alignment. For interpretation, the alignment between vision and language representations can be understood by comparing their semantics with concepts. For enhancement, the alignment can be strengthened by aligning vision-language representations at the concept level, contributing to performance improvements in downstream tasks, including zero-shot image classification and hallucination elimination. Codes are available at https://github.com/ssfgunner/VL-SAE.

## 1   Introduction

Vision-Language Models (VLMs) have demonstrated remarkable capabilities in multi-modal understanding and reasoning, largely attributed to the various training objectives [22, 41] and architectures [57, 27, 66, 65] that effectively align the semantics of vision and language representations. This alignment mechanism serves as a core component of VLMs, enabling them to make predictions by integrating information from both modalities [41, 34, 60, 27, 55, 17]. Nevertheless, current comprehension of the alignment mechanism remains insufficient [13, 49, 38, 2]. This limited comprehension hinders our ability to analyze and address the misalignment cases, such as hallucinations [33, 26, 21]. To tackle this challenge, existing methods interpret representations of VLMs by mapping their semantics to concepts. However, these methods either solely focus on the vision [32, 2] or the language [38] representations, as shown in Figure 1(a). It remains an open problem to interpret the vision-language alignment, which requires not only understanding the representation semantics but also comparing the semantics of both modalities in an interpretable manner [49].

---

*Corresponding author.

39th Conference on Neural Information Processing Systems (NeurIPS 2025).

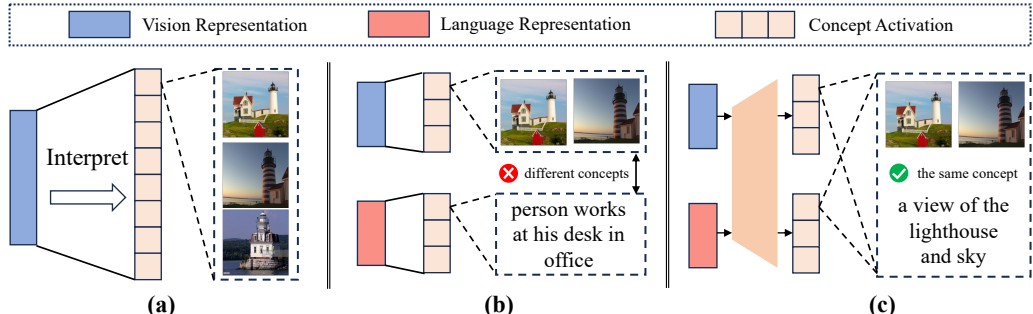

Figure 1: (a) Current interpretation methods are designed for single-modal representations. (b) Using current methods for each modality leads to a mismatch in the concept sets, hindering the interpretation of the vision-language alignment. (c) We propose the VL-SAE to interpret the alignment mechanism by mapping the representation semantics of both modalities into a unified concept set.

A straightforward approach to address this problem is to apply existing methods [46, 8] to map the representation semantics of each modality to concepts separately. With these concepts, we can compare the semantics of vision and language representations for interpreting their alignment mechanism. However, current methods struggle to conduct this comparison with concepts, *as they cannot effectively map the semantics of vision and language representations into a unified concept set*. These methods acquire the concept set in a pre-defined [24, 46, 2] or learnable [35, 8, 32] manner. Pre-defined methods rely on manually constructed concept sets, which often fail to capture the full spectrum of representation semantics [46, 63] and suffer from limited scalability due to the need to collect labeled samples for each concept [24]. Learnable methods employ the hidden activations of a pre-trained sparse autoencoder (SAE) to acquire the concept set, where each hidden neuron correlates to a concept and is defined by the samples that most strongly activate it. The neuron-concept correlation is established via end-to-end learning under self-supervision, eliminating the need for concept-specific annotations [8, 32]. Nevertheless, the end-to-end learning introduces uncontrollability into the concept set. Due to the uncontrollability, applying two separate SAEs to vision and language representations respectively causes the concept mismatch, *i.e.*, neurons at the same location of different SAEs being correlated with different concepts, as illustrated in Figure 1(b). As a result, semantically similar vision and language representations can exhibit inconsistent activations under concept mismatch, hindering the comparisons of representation semantics with SAE.

As the concept mismatch stems from applying separate SAEs to different modalities, using a shared SAE for both modalities seems to be a promising solution. By encouraging semantically similar representations to exhibit consistent activations during self-supervised training, the hidden layer neurons can be co-activated by semantically similar images and texts as shown in Figure 1(c), thereby mitigating the concept mismatch. Unfortunately, current SAEs designed for single-modal scenarios fail to achieve this objective for two primary reasons. *(i)* In contrast to single-modal representations that use the inner product for measuring semantic similarity, multi-modal representations of different VLMs exhibit varying alignment strategies [34, 27, 43], which complicates the measurement of their semantic similarity. *(ii)* Multi-modal representations exhibit modality-specific distributions [50], making it difficult to ensure the activation consistency of semantically similar representations.

In this paper, we propose the VL-SAE, an SAE architecture equipped with an auxiliary autoencoder, to mitigate the concept mismatch. We perform the representation alignment in an explicit form with the auxiliary autoencoder to measure their semantic similarity and then employ VL-SAE to constrain the activation consistency among semantically similar representations. Specifically, *(i)* for the alignment strategy, we propose an auxiliary autoencoder that maps the original representations to its intermediate representations. The autoencoder is optimized via contrastive loss [43] to ensure that vision-language representations with higher semantic similarity exhibit greater cosine similarity between their intermediate representations. *(ii)* For the architecture of VL-SAE, we propose an encoder that activates neurons based on the Euclidean distance between normalized representations and neuron weights. This metric satisfies the triangle inequality and is related to the cosine similarity, which correlates with semantic similarity rather than distributional information. These properties ensure that only semantically similar representations can be close to the same weights and activate corresponding neurons. Additionally, we propose separate decoders to capture the distributional information of each modality for representation reconstruction, thereby preventing the encoder from embedding modality-specific information into neuron activations that leads to concept mismatch.

In experiments, we construct VL-SAE on multiple VLMs, including Contrastive VLMs (CVLMs) [43] that achieve alignment via retrieval tasks and Large VLMs (LVLMs) [34, 1] that rely on question answering tasks. Evaluations of the proposed VL-SAE and current architectures [8, 15] demonstrate that VL-SAE possesses superior ability for mapping the semantics of vision and language representations into a unified concept set. Subsequently, we utilize VL-SAE to interpret and enhance the vision-language alignment mechanism of VLMs. For interpretation, we explain the model prediction by visualizing the activated concepts of vision and language representations during inference. For enhancement, we propose two strategies to improve the performance of CVLM in zero-shot image classification [25, 59, 18] and LVLM in hallucination elimination [30] by enhancing the vision-language alignment at the concept level. The contributions of our work are as follows:

- We propose the VL-SAE, a model to interpret the alignment mechanism of VLMs by mapping the semantics of vision-language representations into a unified concept set.

- We apply VL-SAE on multiple widely-used VLMs and demonstrate the superior quality of the concept set learned by VL-SAE in experiments.

- We show that VL-SAE can be employed to interpret the model prediction by visualizing the activated concepts, enhancing the performance of CVLMs on zero-shot image classification, and eliminating the hallucinations of LVLMs.

## 2    Related Work

**Vision-Language Models.** To handle the vision-language reasoning tasks, researchers have developed multiple VLMs [41, 34, 1, 43, 60, 55, 17] by effectively aligning their vision and language representations. According to the pre-training tasks, these models can be categorized into Contrastive VLMs (CVLMs) [11, 43] that utilize retrieval tasks and Large VLMs (LVLMs) [34, 1, 60, 55, 17] that incorporate Large Language Models (LLMs) [56] for pre-training with question answering tasks. For CVLMs, current models include a vision encoder and a text encoder, applied with contrastive training strategies that encourage semantically similar representations to achieve high inner product values [11]. For LVLMs, existing methods compose the model with a vision encoder, an LLM, and a light-weight connector that maps the output of the vision encoder to input tokens of the LLM. Vision-language representations are implicitly aligned through pre-training with question answering tasks [57, 1]. Despite the various architectures and pre-training paradigms of VLMs, our proposed VL-SAE can interpret and enhance the alignment mechanism of their vision-language representations.

**Representation Interpretation.** Revealing the semantics of representations stands as one of the primary challenges tackled by interpretable machine learning [40]. Previous methods [24, 62, 64, 46] pre-define a concept set and then collect corresponding samples to derive concept vectors in the representation space. The representations are projected onto these vectors for interpretation. These methods are costly due to the requirements for constructing concept sets and collecting relevant samples [24]. Moreover, these methods encounter challenges in modifying model behavior by adjusting corresponding interpretations, as it is difficult to map these interpretations back to the original representations [46]. Recently, Sparse Autoencoder (SAE) [8, 32, 15, 28] has been recognized as an effective method to interpret representations and modify model behaviors through self-supervised learning. Despite achieving advanced performance, current SAEs remain unsuitable for vision-language representations due to the inconsistency between the learned concept sets of both modalities. To address this problem, we propose the VL-SAE with a distance-based encoder and two modality-specific decoders to interpret vision-language representations with a unified concept set.

**Understanding the Internal Mechanism of VLMs.** There has been an increasing interest in investigating the internal mechanisms of VLMs through the lens of multi-modality [9]. Neuron-based methods [16, 44, 37] reveal the existence of multi-modal neurons that translate vision information to corresponding information in text modality. These neurons are located through important scores like gradient [44] and metrics leveraging architecture information [37]. Representation-based methods attempt to interpret the semantics of vision-language representations with concepts. For CVLMs, SpLiCE [2] and TEXTSPAN [14] disentangle the vision representations of CLIP [41] to texts. For LVLMs, Parekh et al. [38] decompose the token representations to vision-language concepts. SAE-V [35] leverages a sparse autoencoder to token representations for efficient data sampling. Our method falls within the representation-based category. In contrast to existing methods that solely focus on

the vision or language representations, our VL-SAE investigates the alignment of vision-language representations by separately mapping their semantics into a unified concept set.

## 3 Methodology

We first introduce the architecture of current VLMs [41, 34, 1] and SAE [8, 32] (Section 3.1). Then, we perform the VLM alignment in an explicit form (Section 3.2). Finally, we propose the VL-SAE to map the semantics of vision-language representations into a unified concept set (Section 3.3).

### 3.1 Preliminaries

**Contrastive Vision Language Models (CVLMs)**. Existing CVLMs [41, 43] typically consist of a vision encoder and a language encoder. Given an input image and text, these encoders independently compute their respective representations. The semantic similarity between the image and text is then estimated through the cosine similarity between their corresponding representations. By collecting a large number of image-text pairs, CVLMs achieve the vision-language alignment by maximizing the representation similarity between semantically similar image-text pairs.

**Large Vision Language Models (LVLMs)**. A general LVLM architecture [34, 27, 1] includes a vision encoder, an LLM, and a connector. An input image is first encoded through the vision encoder and then transformed into image tokens via the connector. Subsequently, these image tokens are concatenated with text tokens and provided as input to the LLM for text generation. With the image-text pairs, the vision-language representations of LVLMs are aligned by pre-training the model to generate textual answers for the text question related to the image content.

To conduct a uniform analysis of both types of models, we utilize the outputs of the vision and language encoders in CVLM for analysis. For LVLM, representations of the image and text tokens in the LLM's hidden layer are averaged across the token axis for analysis. For convenience, we utilize $\mathbf{x}_v, \mathbf{x}_l \in \mathbb{R}^d$ to uniformly represent the vision-language representations of CVLM and LVLM.

**Sparse Autoencoder (SAE)**. SAE is an autoencoder with sparsity constraints on its hidden activations. Given an input $\mathbf{x} \in \mathbb{R}^d$, SAE first transforms it to hidden activations $\mathbf{h} \in \mathbb{R}^h$ using an encoder $E : \mathbb{R}^d \to \mathbb{R}^h$, and then maps it back to the representation space via a decoder $D : \mathbb{R}^h \to \mathbb{R}^d$,

$$\hat{\mathbf{x}} = D(\mathbf{h}) = D(\sigma(E(\mathbf{x}))), \tag{1}$$

where $\sigma : \mathbb{R}^h \to \mathbb{R}^h$ denotes the sparsification function. SAE is trained with the reconstruction loss $\|\hat{\mathbf{x}} - \mathbf{x}\|_2^2$ in a self-supervised manner. For sparsity constraints, previous methods usually select ReLU as the sparsification function $\sigma$ and integrate the $l_1$ norm of $\mathbf{h}$ into the loss function [8]. Recently, researchers have found that directly adopting the top-k operation as the sparsification function not only eliminates the need for an additional loss term but also demonstrates superior scalability [15, 46].

### 3.2 Explicit Representation Alignment

To ensure the activation consistency among semantically similar vision-language representations for alleviating the concept mismatch, a prerequisite is measuring the semantic similarity of representations from both modalities. For semantic similarity measurement, more semantically similar vision-language representations have higher cosine similarity in CVLMs [41, 43], which is explicitly constrained via the pre-training objective. However, it is difficult to measure the semantic similarity of LVLM representations [34, 1] due to their implicit alignment mechanism [50]. To address this problem, we propose employing an auxiliary autoencoder to convert the implicit alignment mechanism into an explicit alignment mechanism. Specifically, given an image-text pair, we input them into the VLM to extract representations $(\mathbf{x}_v, \mathbf{x}_l)$. The autoencoder transforms these representations into intermediate representations $\mathbf{x}_v^e, \mathbf{x}_l^e \in \mathbb{R}^d$ via encoders $\{E_v, E_l\}$ and then maps them back into the original representations $\hat{\mathbf{x}}_v, \hat{\mathbf{x}}_l \in \mathbb{R}^d$ via decoders $\{D_v, D_l\}$,

$$\hat{\mathbf{x}}_v = D_v(\mathbf{x}_v^e) = D_v(E_v(\mathbf{x}_v)), \quad \hat{\mathbf{x}}_l = D_l(\mathbf{x}_l^e) = D_l(E_l(\mathbf{x}_l)). \tag{2}$$

With the contrastive loss InfoNCE [41] and the reconstruction loss, the intermediate representations of image-text pairs can be aligned in cosine similarity while preserving the information of the original representations (details of InfoNCE are provided in Appendix A),

$$\mathcal{L}(\mathbf{x}_v, \mathbf{x}_l) = \texttt{InfoNCE}(\mathbf{x}_v^e, \mathbf{x}_l^e, \mathbf{x}_v^{e-}, \mathbf{x}_l^{e-}) + \|\hat{\mathbf{x}}_v - \mathbf{x}_v\|_2^2 + \|\hat{\mathbf{x}}_l - \mathbf{x}_l\|_2^2, \tag{3}$$

where $\mathbf{x}_v^{e-}, \mathbf{x}_l^{e-}$ denote the intermediate representations of negative samples with dissimilar semantics from the given image-text pair. With this autoencoder, we can transform implicitly aligned representations into explicitly aligned intermediate representations, enabling the measurement of semantic similarity between vision and language representations. Note that this component is only utilized for LVLMs with the implicit alignment mechanism. For CVLMs with the explicit alignment mechanism, we directly use their original representations $\mathbf{x}_v^e = \mathbf{x}_v, \mathbf{x}_l^e = \mathbf{x}_l$ for interpretation.

### 3.3 Interpreting Vision-Language Representations with a Unified Concept Set

Based on explicitly aligned representations, we propose the VL-SAE with a distance-based encoder and two modality-specific decoders to ensure the activation consistency of semantically similar representations, thereby alleviating the concept mismatch and interpreting multi-modal representations with a unified concept set.

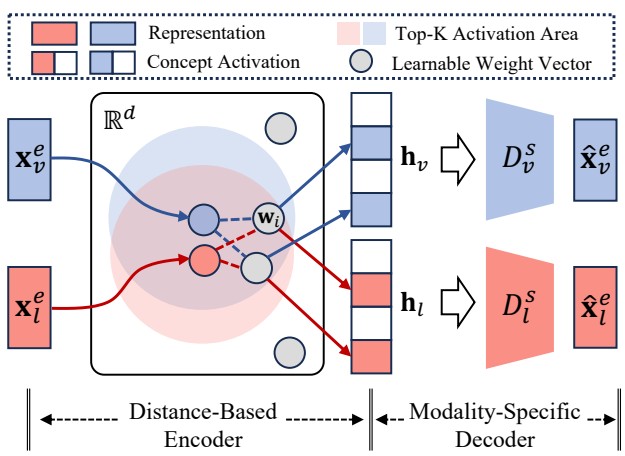

**Encoder**. As the vision-language representations are aligned through cosine similarity, we propose an encoder that activates neurons based on the cosine similarity between the corresponding learnable weight vectors and the input representations. However, since cosine similarity does not satisfy the triangle inequality, two representations with high cosine similarity may not exhibit consistent similarity with the same weight vector, which hinders their activation consistency.

Figure 2: VL-SAE consists of a distance-based encoder that maps semantically similar vision-language representations to similar activations, and two modality-specific decoders that reconstruct original representations from these activations.

To address this problem, we design a distance $g : \mathbb{R}^d \times \mathbb{R}^d \to [0, 2]$ that utilizes the Euclidean distance between normalized inputs. Given the representation $\mathbf{x}^e \in \mathbb{R}^d$ and weight vector $\mathbf{w}_i \in \mathbb{R}^d$ corresponding to the $i$-th hidden neuron, their distance is calculated by

$$g(\mathbf{x}^e, \mathbf{w}_i) = \left\| \frac{\mathbf{x}^e}{\|\mathbf{x}^e\|_2} - \frac{\mathbf{w}_i}{\|\mathbf{w}_i\|_2} \right\|_2 = \sqrt{2 - 2cos(\mathbf{x}^e, \mathbf{w}_i)}, \tag{4}$$

where $cos(\mathbf{x}^e, \mathbf{w}_i)$ denotes the cosine similarity of $\mathbf{x}^e$ and $\mathbf{w}_i$. Since the distance $g$ is a variant of the Euclidean distance, it naturally satisfies the triangle inequality. This indicates that the difference in distance values between the weight vector $\mathbf{w}_i$ and vision-language representations $\mathbf{x}_v^e, \mathbf{x}_l^e$ cannot exceed the distance between the two representations,

$$|g(\mathbf{x}_v^e, \mathbf{w}_i) - g(\mathbf{x}_l^e, \mathbf{w}_i)| \leq g(\mathbf{x}_v^e, \mathbf{x}_l^e). \tag{5}$$

The proposed metric exhibits a negative correlation with cosine similarity, suggesting that as the cosine similarity of vision-language representation increases, the upper bound of the difference in their activations decreases accordingly. We utilize the negative value of $g$ as the activation value, thereby ensuring the positive correlation between the activation value and cosine similarity.

$$E^s(\mathbf{x}^e)[i] = 2 - g(\mathbf{x}^e, \mathbf{w}_i) = 2 - \sqrt{2 - 2cos(\mathbf{x}^e, \mathbf{w}_i)}, \tag{6}$$

where $E^s(\mathbf{x}^e)[i]$ denotes the $i$-th value of $E^s(\mathbf{x}^e) \in \mathbb{R}^h$. We introduce a constant 2 to ensure the non-negativity of the activation values. For the sparsification function $\sigma$, we employ the top-k function (*i.e.,* retaining top-k largest values while setting others to zero) considering its strong scalability [15, 47, 48],

$$\mathbf{h} = \mathtt{TopK}(E^s(\mathbf{x}^e)). \tag{7}$$

The number of activated neurons $k$ serves as a hyper-parameter. This distance-based encoder bounds the activation discrepancy of semantically similar representations through the triangle inequality, promoting each neuron to be co-activated by semantically similar images and texts.

**Decoder.** Despite bounding the activation discrepancy of vision-language representations through the distance-based encoder, it remains insufficient for interpreting representations from distinct distributions. This is because the decoder reconstructs representations solely based on hidden activations. Reconstructing vision-language representations with the same decoder leads to the incorporation of distributional information into the hidden activations during training, thereby reducing the activation consistency of vision-language representations that have similar semantics.

To map the hidden activations into representations of distinct distributions, we propose employing separate decoders for each modality. Given the activations $\mathbf{h}_v, \mathbf{h}_l \in \mathbb{R}^h$ of vision and language representations $\mathbf{x}_v^e, \mathbf{x}_l^e \in \mathbb{R}^d$, two modality-specific decoders $D_v^s, D_l^s$ are utilized to transform these activations back to their original representations, respectively,

$$\hat{\mathbf{x}}_v^e = D_v^s(\mathbf{h}_v), \ \hat{\mathbf{x}}_l^e = D_l^s(\mathbf{h}_l). \tag{8}$$

VL-SAE is trained through the reconstruction loss of visual-language representations,

$$\mathcal{L}(\mathbf{x}_v^e, \mathbf{x}_l^e) = \|\hat{\mathbf{x}}_v^e - \mathbf{x}_v^e\|_2^2 + \|\hat{\mathbf{x}}_l^e - \mathbf{x}_l^e\|_2^2. \tag{9}$$

With the separate decoders for storing distributional information, VL-SAE can encode semantically similar vision-language representations to consistent concept activations as interpretations, and then map these activations back to representations of distinct distributions.

## 4 Experiments

In experiments, we first provide the implementation details for constructing VL-SAE based on VLM representations in Section 4.1. Next, we evaluate the concepts learned by VL-SAE in Section 4.2. Finally, we integrate VL-SAE into the inference process of pre-trained VLMs to interpret and enhance their vision-language alignment mechanisms in Section 4.3.

### 4.1 Implementation Details for VL-SAE Construction.

**VLM Selection.** To demonstrate the broad applicability of the proposed VL-SAE, we build it upon multiple representative VLMs, including CVLMs (OpenCLIP [43] with ViT-B/32, ViT-B/16, ViT-L/14, ViT-H/14) pre-trained with retrieval tasks, and LVLMs (LLaVA1.5 [34], Qwen-VL [1]) pre-trained with question answering tasks. For CVLMs, VL-SAE is constructed using the output from both their vision and language encoders. For LVLMs, VL-SAE is constructed using the output features of the LLM hidden layers (the 29-th layer of LLaVA1.5 and the 26-th layer of Qwen-VL).

**Datasets.** We fed the CC3M dataset [45] containing 3 million image-text pairs into the VLM to extract the corresponding vision-language representations. These representations are randomly divided into training and test sets at a ratio of 4:1.

**Training Strategies.** For LVLMs [34, 1], we first train the auxiliary autoencoder for 50 epochs to perform the explicit representation alignment. The training process is configured with a batch size of 2048, a weight decay of 0.01, and a learning rate of 5e-5. Then, the VL-SAE is trained for 10 epochs based on the intermediate representations of the autoencoder with a batch size of 512 and a learning rate of 1e-4. For CVLMs [43], we adopt the same strategy as the LVLMs but omit the training of the auxiliary autoencoder because CVLM representations are naturally aligned through cosine similarity.

### 4.2 Evaluating the Quality of VL-SAE Concept Set

**Quantitative Evaluation**. For the concept set, we evaluate *(i)* whether each neuron is activated by semantically similar images and texts, and *(ii)* whether diverse semantic concepts are learned across different neurons. These two aspects are quantitatively assessed using the CLIP score [41]. Specifically, we first gather the maximally activating images and texts for each neuron of the VL-SAE's hidden layer to describe its semantics. Then, the CLIP score is employed to quantify the semantic similarity between images and texts within the same neuron (Intra-Similarity) and across different neurons (Inter-Similarity). The Intra-Similarity evaluates the semantic consistency of images and texts within each concept, while the Inter-Similarity assesses the semantic diversity across the concept set. Detailed descriptions of both metrics are presented in Appendix A.1.

For comparison, we consider two baselines using current SAE architectures *(i)* SAE-D, which employs distinct SAEs for vision and language representations, and *(ii)* SAE-S, which uses a single

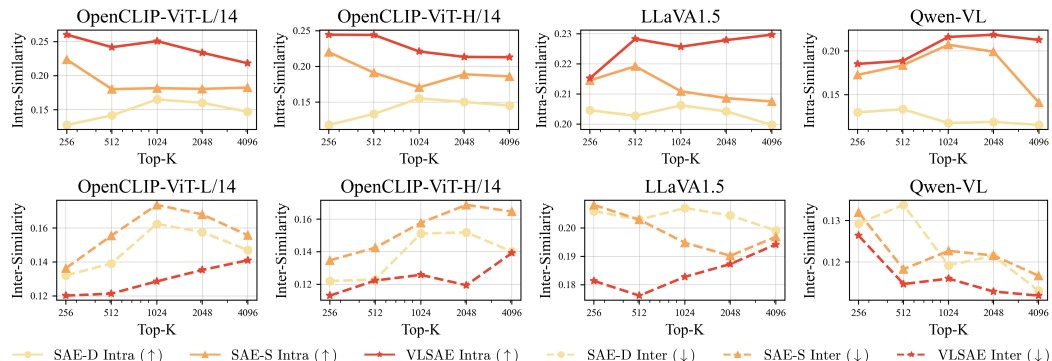

Figure 3: Quantitative evaluation of the learned concept set. We compare the VL-SAE with other methods on the consistency between vision and language semantics within the same neuron (Intra-Similarity) and the semantic diversity across different neurons (Inter-Similarity) on multiple VLMs.

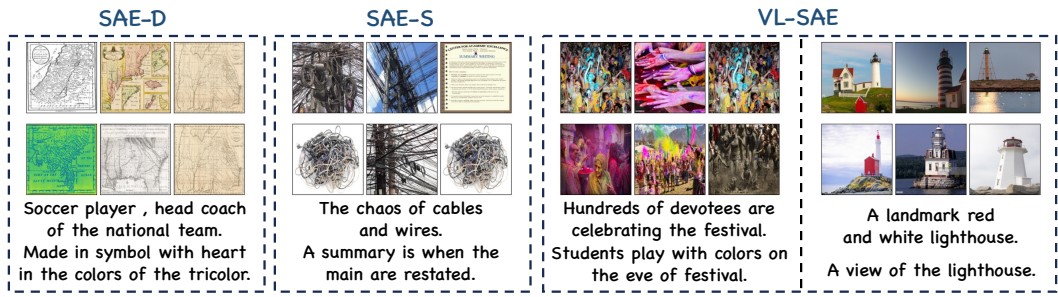

Figure 4: Qualitative comparisons among concepts of SAE-D, SAE-S and the proposed VL-SAE. All SAEs are trained with the vision-language representations of LLaVA 1.5.

SAE shared by both modalities. As shown in Figure 3, VL-SAE exhibits higher intra-similarity and lower inter-similarity compared to these methods, consistently across various VLMs and different numbers of activated concepts. This suggests that *the concepts learned by VL-SAE possess more consistent vision-language semantics and richer semantic diversity than current methods.*

**Qualitative Evaluation.** We provide qualitative comparisons among SAE-D, SAE-S, and our VL-SAE in Figure 4. SAE-D, which employs separate SAEs for each modality, can extract concepts with consistent semantics within the same modality but encounters the concept mismatch issue as illustrated in Figure 1(b). SAE-S, which utilizes a shared SAE for both modalities, demonstrates a less significant concept mismatch compared to SAE-D. However, semantic inconsistencies still exist, such as the conflation of the semantics of "wire" and "summary tutorials" under a single concept. In contrast, the concepts in VL-SAE consist of samples with consistently aligned semantics.

## 4.3 Interpreting and Enhancing the Vision-Language Alignment

With the VL-SAE that correlates the semantics of vision and language representations with a unified concept set, we can interpret and enhance the vision-language alignment mechanism in CVLMs and LVLMs by comparing the semantics of both modalities using the concept set.

**Interpreting Vision-Language Alignment of CVLMs.** Given an image-text pair, we separately map the corresponding vision and language representations into the unified concept set through VL-SAE. Figure 5 depicts several examples by visualizing the concepts activated by representations for each modality and the aligned concepts that are co-activated by both modalities. We observe that *(i)* single-modal representations can activate concepts with irrelevant semantics (*e.g.,* the representations of a motorcycle image activate the concept of a car). This phenomenon reveals the mismatch between representation similarity and semantic similarity in single-modal samples, which arises due to contrastive learning's inability to effectively model the relationships among representations within the same modality. *(ii)* The aligned concepts exhibit strong semantic similarity to both the given image and text, suggesting that the irrelevant concepts activated by a single-modal representation are not simultaneously activated by representations of the other modality.

Figure 5: Interpreting the vision-language alignment of OpenCLIP-ViT-L/14 with VL-SAE. Given an image-text pair, we visualize the concepts activated by corresponding vision and language representations separately, as well as the aligned concepts co-activated by both modalities. Concepts only related to single-modal representations are highlighted in purple.

Table 1: Performance of OpenCLIP with varying model sizes in zero-shot image classification tasks.

| | Caltech101 | Cifar10 | Cifar100 | Country211 | DTD | Eurosat | Flowers | Food | GTSRB | MNIST | Pets | SST2 | STL10 | Sun397 | Mean Acc. |
|---|---|---|---|---|---|---|---|---|---|---|---|---|---|---|---|
| ViT-B/32 | 86.5 | 93.6 | 75.5 | 16.7 | 55.9 | 47.2 | 71.6 | 82.7 | 49.3 | 69.8 | 90.7 | 57.5 | 96.6 | 68.7 | 68.7 |
| +VL-SAE | 86.7 | 93.8 | 75.7 | 16.7 | 56.9 | 51.0 | 71.8 | 82.8 | 50.3 | 71.8 | 90.7 | 57.9 | 96.9 | 69.4 | **69.5** |
| ViT-B/16 | 86.7 | 94.9 | 76.9 | 20.3 | 56.5 | 52.6 | 71.4 | 86.6 | 46.1 | 66.0 | 90.3 | 59.7 | 97.9 | 70.8 | 69.8 |
| +VL-SAE | 87.2 | 95.1 | 77.0 | 20.4 | 56.7 | 53.2 | 71.6 | 86.8 | 49.1 | 67.8 | 90.9 | 60.4 | 97.9 | 71.4 | **70.4** |
| ViT-L/14 | 87.6 | 95.8 | 78.3 | 24.4 | 61.5 | 57.8 | 74.4 | 88.8 | 51.6 | 64.5 | 92.9 | 60.3 | 98.5 | 74.0 | 72.2 |
| +VL-SAE | 87.9 | 95.9 | 78.6 | 24.5 | 62.8 | 60.3 | 74.5 | 88.8 | 51.9 | 69.1 | 93.0 | 60.6 | 98.6 | 74.3 | **72.9** |
| ViT-H/14 | 88.2 | 97.5 | 84.7 | 29.8 | 67.9 | 72.7 | 80.1 | 92.7 | 58.3 | 72.9 | 94.5 | 64.1 | 98.5 | 75.2 | 76.9 |
| +VL-SAE | 88.5 | 97.7 | 85.0 | 30.0 | 68.4 | 76.7 | 80.1 | 92.9 | 59.5 | 76.1 | 94.7 | 65.2 | 98.6 | 75.3 | **77.8** |

**Enhancing Vision-Language Alignment of CVLMs.** According to the above interpretations, mapping vision-language representations to a unified concept set enables filtering out the irrelevant concepts activated by single-modal representations. Inspired by this, we enhance the vision-language alignment of CVLMs by aligning their multi-modal representations in concepts. Specifically, for an input image $x_v$ and text $x_l$, their semantic similarity depends not only on the cosine similarities of their representations $\mathbf{x}_v, \mathbf{x}_l$, but also on the cosine similarities of their concept activations $\mathbf{h}_v, \mathbf{h}_l$.

$$y = cos(\mathbf{x}_v, \mathbf{x}_l) + \alpha_c cos(\mathbf{h}_v, \mathbf{h}_l), \tag{10}$$

where $\alpha_c$ is a task-specific hyperparameter that controls the proportion of concept-based predictions. We enhance the alignment mechanism of multiple CVLMs and conduct evaluations across various zero-shot image classification datasets [59, 25, 41, 6, 18, 36, 3, 19, 10, 39, 51, 7, 61]. As Table 1 shows, mapping multi-modal representations to a unified concept set for predictions achieves consistent performance improvement across different models and datasets.

**Interpreting Vision-Language Alignment of LVLMs.** For LVLMs, we interpret their alignment mechanism by visualizing the concepts activated by the vision and language representations during the text generation process. As Figure 6 shows, given the input image of a kitchen including ovens and stoves, the generated text exhibits object hallucinations of the microwave and refrigerator. By analyzing the activated concepts, we observe that the semantics of both vision and language representations are correlated with the kitchen scene but emphasize different objects. Specifically, the vision representation focuses on ovens and stoves, whereas the language representation is associated with microwaves and refrigerators, which correspond to the hallucinated objects. This phenomenon indicates that the object information in the vision representation is inadequately transferred into the language representation, resulting in hallucinations in the generated text.

**Enhancing Vision-Language Alignment of LVLMs.** As the hallucinations of LVLMs are correlated to the misalignment between vision and language representations, we leverage VL-SAE to mitigate

| Input Image | Concepts activated by image embeddings | Generated Text | Concepts activated by text embeddings |
|---|---|---|---|

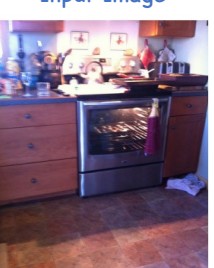

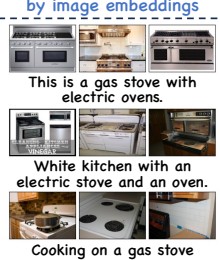

This is a gas stove with electric ovens.

White kitchen with an electric stove and an oven.

Cooking on a gas stove in the kitchen

The image features a kitchen with a stainless steel oven and a stove top positioned next to each other. In addition to the oven and stove top, the kitchen also contains a sink, a microwave, and a refrigerator. There are several bottles placed around the kitchen, with some on the countertops and others on the floor. A cup can be seen on the countertop as well. The kitchen appears to be well-equipped and ready for cooking.

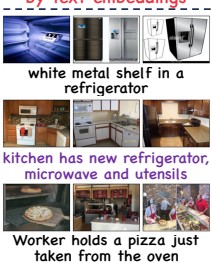

white metal shelf in a refrigerator

kitchen has new refrigerator, microwave and utensils

Worker holds a pizza just taken from the oven

Figure 6: Interpreting the vision-language alignment of LLaVA1.5 with VL-SAE. Given an input image and the text generated by LVLMs, we visualize the concepts activated by corresponding vision and language representations. The hallucination concept activated by language representations facilitates a deeper understanding of object hallucinations in the text.

Table 2: Experimental results on POPE for hallucination elimination.

| Setting | Decoding | LLaVA1.5 | | | | Qwen-VL | | | |
|---|---|---|---|---|---|---|---|---|---|
| | | Accuracy | Precision | Recall | F1 Score | Accuracy | Precision | Recall | F1 Score |
| Random | Regular | 82.93 | 92.01 | 72.13 | 80.87 | 85.20 | 95.99 | 73.47 | 83.23 |
| | VCD | 85.53 | 93.68 | 76.20 | 84.04 | 86.33 | 96.11 | **75.73** | 84.71 |
| | VL-SAE | **87.07** | **97.28** | **76.27** | **85.50** | **86.47** | **97.15** | 75.13 | **84.74** |
| Popular | Regular | 81.13 | 87.96 | 72.13 | 79.27 | 83.80 | 94.47 | 71.80 | 81.59 |
| | VCD | 83.63 | 89.51 | 76.20 | 82.31 | 85.73 | 94.30 | **76.10** | 84.21 |
| | VL-SAE | **85.87** | **94.39** | **76.27** | **84.37** | **85.97** | **96.15** | 74.93 | **84.22** |
| Adversarial | Regular | 78.67 | 83.03 | 72.07 | 77.16 | 82.33 | 89.88 | 72.87 | 80.49 |
| | VCD | 81.10 | 84.47 | **76.20** | 80.13 | 83.77 | 90.03 | **75.93** | 82.39 |
| | VL-SAE | **83.60** | **89.44** | **76.20** | **82.29** | **86.10** | **97.29** | 74.26 | **84.23** |

hallucinations by enhancing the vision-language alignment at the concept level. Specifically, with the vision and language representations $\mathbf{x}_v, \mathbf{x}_l$, we obtain a refined language representation $\hat{\mathbf{x}}_l$ by aligning its concept activation $\mathbf{h}_l$ with vision concept activation $\mathbf{h}_v$ through VL-SAE,

$$\hat{\mathbf{x}}_l = (1 - \alpha_l)\mathbf{x}_l + \alpha_l D_l(D_l^s(\mathbf{h}_l + \beta\mathbf{h}_v)). \tag{11}$$

The modified language representation $\hat{\mathbf{x}}_l$ is utilized to mitigate hallucinations through contrastive decoding [26]. More implementation details are provided in Appendix A. Table 2 shows the performance comparisons on the POPE [30] benchmark. As a method for representation interpretation, VL-SAE surpasses the VCD [26] that specializes in hallucination elimination. The effectiveness of VL-SAE further highlights the potential of representation interpretation methods to improve model performance on downstream tasks by enhancing the vision-language alignment.

### 4.4 Ablation Studies & Visualizations

**Effects of the VL-SAE Architecture**. The ablation studies of the proposed components are provided in Table 3. First, replacing the standard encoder with the cosine-based encoder (+Cosine-based Encoder), which activates hidden neurons based on cosine similarity, can improve the quality of learned concepts, as vision-language representations are explicitly aligned through cosine similarity.

Moreover, replacing the cosine similarity with the distance proposed in Equation 4 can improve the concept quality (+Distance-based Encoder). This is because the distance satisfies the triangle inequality, enabling each hidden neuron to be activated by more semantically similar visual-language representations.

Table 3: Ablation studies of the proposed architecture.

| | OpenCLIP-ViT-H/14 | | LLaVA1.5 | |
|---|---|---|---|---|
| | Intra-Sim. ($\uparrow$) | Inter-Sim. ($\downarrow$) | Intra-Sim. ($\uparrow$) | Inter-Sim. ($\downarrow$) |
| Standard SAE | 0.1890 | 0.1688 | 0.2086 | 0.1902 |
| +Auxiliary Autoencoder | - | - | 0.2092 | 0.1908 |
| +Cosine-based Encoder | 0.1891 | 0.1518 | 0.2103 | 0.1902 |
| +Distance-based Encoder | 0.2016 | 0.1357 | 0.2216 | 0.1842 |
| +Modality-specific Decoder | **0.2134** | **0.1149** | **0.2257** | **0.1828** |
| -Auxiliary Autoencoder | - | - | 0.2084 | 0.2034 |

Additionally, the improvement achieved by adopting modality-

Table 4: Concept quality of VL-SAE pre-trained with different proportions of the CC3M dataset.

| Data Percentage | 10% | 30% | 50% | 70% | 100% |
|---|---|---|---|---|---|
| Intra-Similarity ($\uparrow$) | 0.2029 | 0.2129 | 0.2196 | 0.2222 | **0.2299** |
| Inter-Similarity ($\downarrow$) | 0.1597 | 0.1306 | 0.126 | 0.1256 | **0.1220** |

Table 5: Ablation studies of the sparsification method in VL-SAE.

| Method | Intra-Similarity ($\uparrow$) | Inter-Similarity ($\downarrow$) |
|---|---|---|
| L1 | 0.2142 | 0.1809 |
| Top-K | **0.2442** | **0.1373** |

**Concept 2153**

Classic and romantic photograph of the couple. Bouquet by person at this beautiful wedding.

**Concept 5478**

Summer vacation – young woman lying on the beach. Sunny summer at the beach with palm trees and glasses.

**Concept 6169**

Baseball player hits a–run home run. Baseball player hits a run – scoring single.

**Concept 7301**

A banner or poster for happy new year. A background for happy new year.

Figure 7: Qualitative evaluation of the learned concept set. We provide the maximally activating images and texts for several concepts of VL-SAE trained with the representations of LLaVA 1.5.

specific decoders stems from mitigating the adverse influence of distributional discrepancies in multi-modal representations on the encoder. Furthermore, directly training the VL-SAE with original representations of LVLMs (-Auxiliary Autoencoder) causes the failure of other components, highlighting the necessity of transforming the alignment mechanism to an explicit form.

**Effects of the Dataset Volume**. In Table 4, we train VL-SAE models with different proportions of the CC3M dataset and report their Intra-Similarity and Inter-Similarity. As the volume of data increases, the Intra-Similarity (0.2029→0.2299) and Inter-Similarity (0.1597→0.1220) of the concepts learned by VL-SAE improve accordingly. This suggests that VL-SAE can learn higher-quality concepts by leveraging larger amounts of data in a self-supervised manner.

**Effects of the Sparsification Method**. In Table 5, we compare the performance of utilizing Top-k [47, 46] and L1 loss [8] for sparsification. The coefficient of L1 loss is set to 1e-4. We find that employing Top-K sparsification yields superior concept quality compared to L1 loss, as the latter lacks precise control over the number of activated hidden neurons. This limitation hinders its ability to effectively balance sparsity in neuron activation with the self-supervised representation reconstruction loss. Our results are consistent with previous studies [15], which also indicate that Top-K sparsification is a more effective and scalable approach.

**Visualizations of Concepts Learned by VL-SAE**. In Figure 7, we visualize the learned concepts of VL-SAE, which are represented by their maximally activating images and texts. The images and texts of the same concept exhibit highly similar semantics, demonstrating that VL-SAE can effectively alleviate the concept mismatch. Additionally, VL-SAE obtains concepts with abstract semantics rather than focusing exclusively on semantics related to visual appearance. For example, concept 5478 is style-agnostic and activated by both natural and cartoon images, which depict humans lying on a beach. Concept 7301 is color-agnostic and activated by images of happy new year posters.

## 5 Conclusion

In this work, we map the semantics of vision-language representations into a unified concept set with VL-SAE, which is trained by encouraging the consistency of its hidden activations among semantically similar representations under self-supervision. To measure the semantic similarity of multi-modal representations, we perform their alignment in an explicit form that correlates the semantic similarity with cosine similarity. To ensure the activation consistency among semantically similar representations, we propose the VL-SAE architecture, including a distance-based encoder and two modality-specific decoders. Experimental results demonstrate the superior ability of VL-SAE to interpret and enhance the alignment mechanism of VLMs. For future work, we will apply the proposed method to more VLMs and investigate its capabilities in additional tasks such as model unlearning and continuous learning. For limitations discussion, please refer to Appendix C.4.

## Acknowledgement and Disclosure of Funding

This work was supported in part by the National Key R&D Program of China under Grant 2023YFC2508704, in part by the National Natural Science Foundation of China 62236008, and in part by the Fundamental Research Funds for the Central Universities. The authors would like to thank Siqi Zhang, Yue Wu, and the anonymous reviewers for their constructive comments and suggestions that improved this manuscript.

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

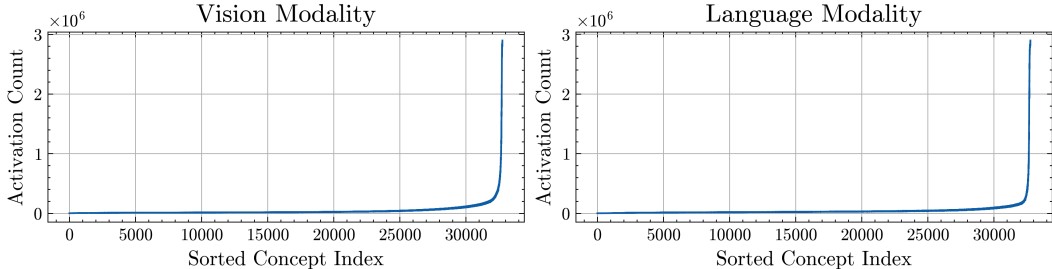

Figure A8: Activation count of each concept in VL-SAE constructed for LLaVA1.5 [34].

Table A6: The $\alpha$ value of OpenCLIP with varying model sizes in zero-shot image classification tasks.

| | Caltech101 | Cifar10 | Cifar100 | Country211 | DTD | Eurosat | Flowers | Food | GTSRB | MNIST | Pets | SST2 | STL10 | Sun397 |
|---|---|---|---|---|---|---|---|---|---|---|---|---|---|---|
| ViT-B/32 | 0.6 | 0.6 | 0.3 | 0.1 | 0.1 | 0.2 | 0.1 | 0.5 | 0.4 | 1.0 | 0.1 | 0.3 | 0.1 | 0.2 |
| ViT-B/16 | 0.6 | 0.5 | 0.3 | 0.1 | 0.1 | 0.1 | 0.1 | 0.5 | 0.5 | 1.0 | 0.1 | 0.2 | 0.1 | 0.2 |
| ViT-L/14 | 0.6 | 0.4 | 0.3 | 0.3 | 0.1 | 0.3 | 0.2 | 0.1 | 0.2 | 1.0 | 0.1 | 0.1 | 0.9 | 0.1 |
| ViT-H/14 | 0.4 | 0.8 | 0.6 | 0.1 | 0.3 | 0.7 | 0.1 | 0.9 | 0.5 | 1.0 | 0.1 | 0.5 | 0.2 | 0.1 |

# A   More Implementation Details

## A.1   Details of the VL-SAE Construction and Evaluation

For the InfoNCE loss utilized in Section 3.2, given an image-text pair $(\mathbf{x}_v^e, \mathbf{x}_l^e)$ and other $N$ with intermediate representations $\{(\mathbf{x}_{v,1}^e, \mathbf{x}_{l,1}^e), ..., (\mathbf{x}_{v,N}^e, \mathbf{x}_{l,N}^e)\}$ which denote the $(\mathbf{x}_v^{e-}, \mathbf{x}_l^{e-})$ in Equation 3, the InfoNCE loss can be represented by

$$\texttt{InfoNCE}(\mathbf{x}_v^e, \mathbf{x}_l^e, \mathbf{x}_v^{e-}, \mathbf{x}_l^{e-}) = -\log\left(\frac{\exp(\frac{cos(\theta_{\mathbf{x}_v^e, \mathbf{x}_l^e})}{\tau})}{\sum_{i=1}^N \exp(\frac{cos(\theta_{\mathbf{x}_v^e, \mathbf{x}_{l,i}^e})}{\tau})}\right) - \log\left(\frac{\exp(\frac{cos(\theta_{\mathbf{x}_v^e, \mathbf{x}_l^e})}{\tau})}{\sum_{i=1}^N \exp(\frac{cos(\theta_{\mathbf{x}_{v,i}^e, \mathbf{x}_l^e})}{\tau})}\right),$$

(A12)

where $\tau$ denotes the temperature hyperparameter, which is set to 0.07. The InfoNCE loss is computed for image-text pairs within the same batch and aggregated to train the auxiliary autoencoder.

For the construction of the VL-SAE and auxiliary autoencoder, all the modules except for the distance-based encoder are implemented with a linear layer. We set the hidden ratio to 8 in both CVLMs and LVLMs, indicating that the number of neurons in the hidden layer is 8 times the representation dimension. Unless otherwise specified, the number of activated neurons for each representation is set to 256 for all CVLMs and LVLMs. For the quantitative evaluation of the concept set, we randomly select several concepts from the autoencoder to compute their intra-similarity and inter-similarity. We calculate these metrics through five random trials, and the average value is adopted to be presented in Figure 3. The impact of the number of concepts used for evaluation is shown in Table A7. The metrics showed no significant differences when more than 100 concepts are used. Since the evaluation needs to be repeated multiple times, we select 100 concepts in each iteration for efficient evaluation.

For the evaluation metrics in Figure 3, for the $i$-th neuron, we first obtain its maximally activating images and texts. Then, these images and texts are transformed into representations with a pre-trained OpenCLIP-ViT-H/14 [43]. The averaged image and text representations $\mathbf{x}_v^i, \mathbf{x}_l^i$ are utilized to compute the Inter-Similarity and Inter-Similarity metrics. The Inter-Similarity is computed as the average cosine similarity between the image and text representations within the same neuron.

$$Sim_{intra} = \frac{1}{h}\sum_{i=1}^h cos(\theta_{\mathbf{x}_v^i, \mathbf{x}_l^i}).$$

(A13)

Table A7: Impact of the number of concepts on the evaluation metrics based on OpenCLIP-ViT-B-16.

| Concept Num. | 100 | 200 | 300 | 400 | 500 |
|---|---|---|---|---|---|
| Intra Similarity (SAE-S) | 0.2200 | 0.2257 | 0.2281 | 0.2245 | 0.2247 |
| Intra Similarity (VL-SAE) | **0.2445** | **0.2508** | **0.2489** | **0.2476** | **0.2469** |
| Inter Similarity (SAE-S) | 0.1347 | 0.1293 | 0.1300 | 0.1306 | 0.1308 |
| Inter Similarity (VL-SAE) | **0.1230** | **0.1214** | **0.1182** | **0.1181** | **0.1195** |

Table A8: Number of concepts within different SAEs.

| Method | Hidden Neuron | Dead Neuron | Concept Num. |
|---|---|---|---|
| SAE-D | 32768 (4096×8) | 54 | 32714 |
| SAE-S | 32768 (4096×8) | 46 | 32722 |
| VL-SAE | 32768 (4096×8) | 15 | 32753 |

For the Inter-Similarity, it is computed as the average cosine similarity between the image and text representations within the different neurons.

$$Sim_{inter} = \frac{1}{h(h-1)} \sum_{i=1}^{h} \sum_{j \neq i}^{h} cos(\theta_{\mathbf{x}_v^i, \mathbf{x}_l^j}). \tag{A14}$$

For the resource requirements, the VL-SAE models for different VLMs [43, 34, 1] are all trained using a single NVIDIA GeForce RTX 4090 GPU. For CVLM [43], full-precision training is employed to ensure optimal performance and stability. For LVLM [34, 1], half-precision (FP16) training is utilized to effectively reduce resource requirements while maintaining model efficiency.

## A.2 Number of Concepts in VL-SAE

In Table A8, we provide the number of concepts in different SAE architectures based on LLaVA 1.5 with 4096 feature dimensions. The number of concepts is upper bounded by the number of hidden neurons in SAEs. Moreover, the number of concepts is influenced by the training process and SAE architectures. This stems from the inevitable existence of dead neurons in SAEs [15], which are never activated. The results show that VL-SAE learns more concepts compared to existing methods.

## A.3 Details of Interpreting Representations with VL-SAE.

We observe that a subset of concepts in VL-SAE are activated at high frequencies, as Figure A8 shown, *i.e.*, almost all samples activate these concepts. This frequent activation makes it challenging for these concepts to be associated with specific semantics, as shown in Tale A9. Consequently, in the interpretation process, we divide the target representation's neuron activation by the average activation value of each neuron, thereby mitigating the impact of these high-frequency concepts on the interpretation.

Table A9: The concepts of VL-SAE relevant to a kitchen image with and without re-weighting. All concepts are shown in text form. The concept corresponding to the high-frequency neuron is denoted by (*), which exhibits irrelevant semantics with the input kitchen image.

| Before re-weighting | After re-weighting |
|---|---|
| toast popping out of a toaster | toast popping out of a toaster |
| hard rock artist performs (*) | burning on a gas stove in the kitchen |
| burning on a gas stove in the kitchen | cans in a refrigerator in a restaurant |
| cans in a refrigerator in a restaurant | this is a gas stove with electric ovens |
| this is a gas stove with electric ovens | gas fueled rings on a domestic cooker |

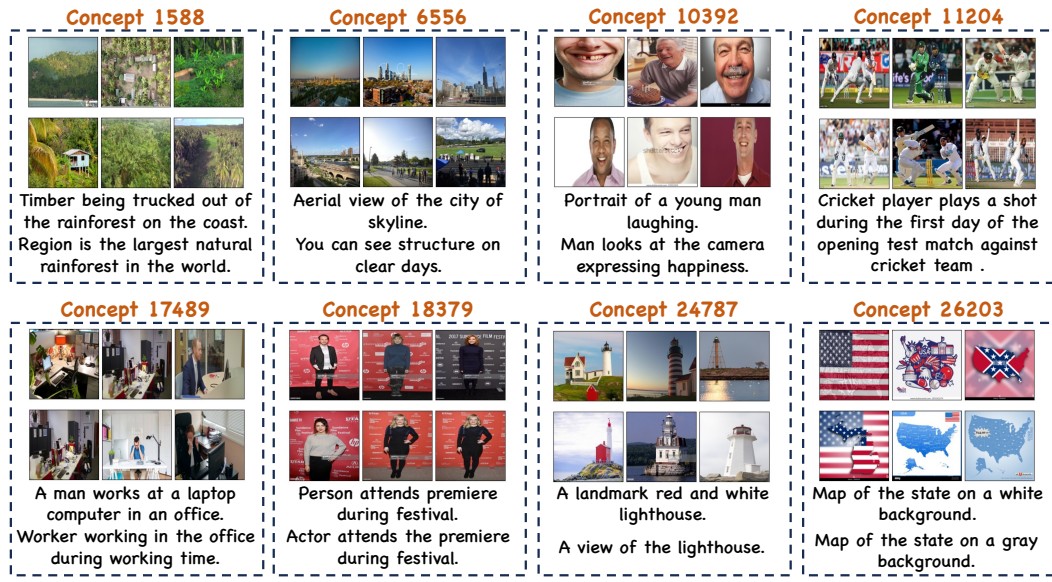

Figure A9: More concepts learned by VL-SAE. We provide the maximally activating images and texts for several concepts of VL-SAE trained for LLaVA1.5.

## A.4  Details of the Zero-shot Image Classification Task

Considering the varying degrees of alignment between the concept set of VL-SAE and the target categories of the downstream task, we experiment with multiple values of $\alpha \in \{0.1, 0.2, 0.3, 0.4, 0.5, 0.6, 0.7, 0.8, 0.9\}$ and select the one that performs best. The selected values are presented in Table A6. We observed that models of varying sizes tended to select different $\alpha$ values, whereas for certain datasets like MNIST and Pets, the optimal alpha values remained consistent across different models. Moreover, we observe that the optimal alpha values for some datasets, such as CIFAR-10 and Caltech101, are significantly higher than those for other datasets. This discrepancy stems from the strong correlation between their target categories and the dataset utilized to train the VL-SAE model.

## A.5  Details of the Hallucination Elimination Task

With the modified representation $\hat{\mathbf{x}}_l$ in Equation 11, we adopt the contrastive decoding strategy [26] to mitigate the hallucination in LVLMs. Specifically, with the representations of vision and language tokens $\mathbf{x}_{vt} \in \mathbb{R}^{N_v \times d}, \mathbf{x}_{lt} \in \mathbb{R}^{N_l \times d}$, the model generates two distinct output distributions: *(i)* one conditioned on the original representations; and *(ii)* the other conditioned on the original vision representations and the language representations with their mean value across the token dimension replaced by $\hat{\mathbf{x}}_l$.

$$\hat{\mathbf{x}}_{lt}[i] = \mathbf{x}_{lt}[i] - \mathbf{x}_l + \hat{\mathbf{x}}_l, \tag{A15}$$

where $\mathbf{x}_{lt}[i] \in \mathbb{R}^d$ denotes the representation of the $i$-th text token, $\mathbf{x}_l \in \mathbb{R}^d$ denotes the language representations averaged across the token dimension. Then, a new contrastive probability distribution is computed by exploiting the differences between the two initially obtained distributions. The new contrastive distribution is formulated as

$$p_{cd}(y \mid \mathbf{x}_{vt}, \mathbf{x}_{lt}, \hat{\mathbf{x}}_{lt}) = \mathrm{softmax}[(1 - \alpha_{cd}) \mathrm{logit}_\theta(y \mid \mathbf{x}_{vt}, \mathbf{x}_{lt}) + \alpha_{cd} \mathrm{logit}_\theta(y \mid \mathbf{x}_{vt}, \hat{\mathbf{x}}_{lt})]. \tag{A16}$$

Larger $\alpha_{cd}$ values indicate a stronger amplification of differences between the two distributions. Moreover, we also utilize the adaptive plausibility constraints [29] that conduct contrastive decoding contingent upon the confidence level $\beta_{cd}$ associated with the output distribution with original representations. In practice, we set $\alpha_{cd}$ and $\beta_{cd}$ to 0.6 and 0.8, respectively. Additionally, the values of $\alpha$ and $\beta$ in Equation 11 are determined to be 0.7 and 0.9, respectively.

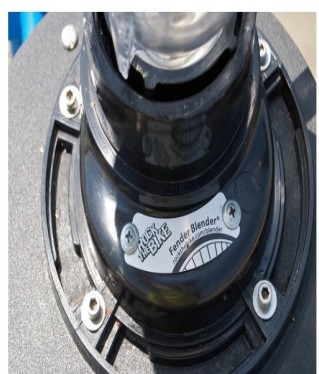

**Regular:**

The image features a smiley face drawn on the inside shiny surface of a black part, either including a metal or a plastic part. There is a gap, or a spotlight, between the metal or plastic part and it appears to be sitting over a motor. The face looks out of place in such an environment, adding a playful touch to the scene.

**With VL-SAE:**

The image features a close-up view of a black object, possibly a wheel or a piece of machinery, with a sticker on it. The sticker has a picture of a smiling face and the words "Fender Blender" written below it. The face appears to be a cartoon-like character, adding a playful touch to the object.

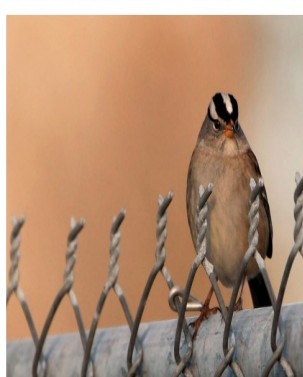

**Regular:**

The image shows a gray bird perched on top of a metal fence. The bird is the main focus of the scene, and it appears to be looking carefully at something in the distance. Near the bird, there is another bird sitting farther down the fence. The entire scene conveys a close-up shot of the birds on the fence.

**With VL-SAE:**

The image shows a small bird perched on top of a metal fence. The bird is standing on the fence, looking at the camera. The fence is made of metal bars, and the bird is positioned in the middle of the fence. The fence is composed of several metal bars, creating a stricter and more secure barrier. The scene captures the bird's curiosity and its interaction with the fence.

Figure A10: Illustration of generated captions by different decoding methods with LLaVA 1.5 as the backbone model. Hallucinated content is highlighted in purple.

Table A10: Human evaluation of concepts learned by different SAEs.

| Method | OpenCLIP-ViT/H | LLaVA 1.5 |
|--------|----------------|-----------|
| SAE-D  | 0.6            | 0.1       |
| SAE-S  | 33.5           | 35.8      |
| VL-SAE | 65.9           | 64.1      |

## B    More Experimental Results

### B.1    Human Evaluation

We randomly select 100 concepts from SAE-D, SAE-S, and VL-SAE, respectively, and invite 10 participants to evaluate these concepts. For each iteration, participants are required to evaluate the quality of three given concepts and select the one with the highest quality. The method corresponding to the selected concept is awarded one point. The concept quality of these methods is measured by the average score assigned by all participants. As Table A10 shows, concepts learned by VL-SAE possess higher quality than those learned by other methods. This comparison further highlights the effectiveness of our proposed VL-SAE.

### B.2    More Results for Hallucination Elimination

Beyond the "Yes-or-No" discriminative evaluations on the POPE datasets, we evaluate the VL-SAE on open-ended caption generation using the CHAIR benchmark [42]. We randomly select 500 samples from CHAIR for evaluation and calculate the average of the results obtained from five independent trials. The results in Table A11 show the superior performance of VL-SAE over the compared methods. Specifically, VL-SAE reduces object hallucinations in generated captions, as evidenced by lower $CHAIR_S$ and $CHAIR_I$ scores. In addition, VL-SAE enhances the detail of the generated captions, as indicated by higher Recall scores.

Table A11: Results on the subset of CHAIR.

| Model | Decoding | CHAIR$_S$ ($\downarrow$) | CHAIR$_I$ ($\downarrow$) | Recall ($\uparrow$) | Avg. Len |
|---|---|---|---|---|---|
| LLaVA1.5 | Regular | 53.4 | 17.6 | 72.3 | 103.2 |
| | VCD | 55.0 | 16.3 | 76.0 | 102.5 |
| | VL-SAE | **47.8** | **13.3** | **76.3** | 100.7 |
| Qwen-VL | Regular | 44.6 | 16.1 | 60.7 | 97.2 |
| | VCD | 42.6 | 13.9 | 60.0 | 94.1 |
| | VL-SAE | **39.6** | **10.7** | **63.3** | 94.0 |

Table A12: Training cost of VL-SAE based on representations of OpenCLIP.

| Base Model | ViT-B/16 | ViT-L/14 | ViT-H/14 |
|---|---|---|---|
| FLOPs | 0.03G | 0.06G | 0.10G |
| Training Time | 132s | 228s | 446s |

## B.3 Resource Requirements of VL-SAE

**Training Costs**. Table A12 presents the training cost of VL-SAE. First, as the model size increases, the dimensionality of its learned representations grows, leading to higher computational demands for training VL-SAE. Second, despite this scaling trend, the overall training cost of VL-SAE remains remarkably low. For instance, when compared to LoRA [20], a widely adopted method known for its computational efficiency, training VL-SAE on OpenCLIP-ViT-B/16 representations incurs significantly fewer FLOPs (0.03G v.s. 1.91G). This minimal computational overhead stems from the small parameter volume in VL-SAE, which is approximately equivalent to that of two linear layers.

**Inference Cost**. Table A13 shows the parameters and computation cost of VL-SAE. For inference speed, we present the influences of VL-SAE in Table A14. The results indicate that VL-SAE has a negligible impact. This is because VL-SAE comprises only two linear layers with significantly smaller parameters than pre-trained VLMs.

## B.4 Additional Ablation Studies

**Selection of Hyperparameter $\alpha_c$.** We first provide accuracy comparisons under three settings: baseline ($\alpha_c = 0$), task-agnostic ($\alpha_c = 0.1$), and task-specific ($\alpha_c$ determined by tasks). The results are shown in Table A15. First, the task-agnostic scheme performs better than the baseline (72.4% v.s. 72.2% on ViT-L, and 77.2% v.s. 76.9% on ViT-H). This indicates that $\alpha$ does not need to be task-specific for performance improvement and can generalize across tasks. However, these tasks originate from different domains and exhibit distinct preferences for hyperparameters [31], as evidenced by the best performance of the task-specific scheme. If the objective is to achieve optimal performance, the $\alpha$ values of each task should be determined individually, To determine the $\alpha$ value of each task, we conduct experiments with multiple values $\alpha \in [0, 1]$ at intervals of 0.1 and select the one that performs the best. Note that the cost of conducting multiple experiments is very low. For instance, when using the Food-101 dataset, each experiment on a single NVIDIA 4090 GPU takes only approximately 90 seconds.

**Selection of Hyperparameter $\alpha_l$.** For the hyperparameter $\alpha_l$ in Equation 11, We conduct experiments by selecting values at intervals of 0.1 within the range of $[0.5 - 0.9]$, and the one with the best performance is selected. We report the performance under different values in Table A16. The highest performance across different settings is achieved at $\alpha_l = 0.7$.

Table A13: Additional parameter and computation cost of VL-SAE.

| Model | Total Param. | VL-SAE Param. | Total FLOPs | VL-SAE FLOPs |
|---|---|---|---|---|
| LLaVA-1.5 | 7529M | 467M | 14.03G | 0.875G |
| OpenCLIP-ViT-H | 1011M | 25M | 3.15G | 0.034G |

Table A14: Throughput (sample/s) of the original models and those equipped with VL-SAE.

| Method | OpenCLIP-ViT-B | OpenCLIP-ViT-L | OpenCLIP-ViT-H | LLaVA 1.5 |
|---|---|---|---|---|
| Original | 935.32 | 213.62 | 99.45 | 11.34 |
| +VL-SAE | 935.13 | 213.42 | 99.29 | 10.69 |

Table A15: Ablation of hyperparameter $\alpha_c$ in Equation 10.

| Method | OpenCLIP-ViT-L | OpenCLIP-ViT-H |
|---|---|---|
| baseline ($\alpha_c = 0$) | 72.2 | 76.9 |
| task-agnostic ($\alpha_c = 0.1$) | 72.4 | 77.2 |
| task-specific ($\alpha_c$ determined by tasks) | **72.9** | **77.8** |

**CC3M training**. In the training process of VL-SAE, the CC3M dataset is employed to integrate VL-SAE as a general plug-in for VLMs, rather than to augment the intrinsic capabilities of these VLMs. In our training strategies, the VLM parameters remain frozen without any adjustments. To further evaluate the impact of CC3M dataset, we provide comparisons between the base model (CLIP-ViT-B/16), the base model fine-tuned by CC3M, and the base model equipped with VL-SAE in Table A17. The results show that fine-tuning does not lead to improved performance on downstream tasks. This is because CC3M serves as a subset of the VLM pre-training dataset. It does not provide the base model with any additional task-relevant information. The superior performance of VL-SAE compared to the fine-tuning model indicates that the improvement stems from the mapping from representations to concepts, rather than the incorporation of CC3M.

## B.5 Additional Visualizations

### B.5.1 Concepts Learned by VL-SAE

We provide more visualization of the concepts learned by VL-SAE in Figure A9. These results highlight the superior capability of VL-SAE in acquiring a vision-language concept set with rich semantics, such as natural landscapes, urban environments, human expressions, and maps.

### B.5.2 Cases for Hallucination Elimination

We present qualitative examples of different decoding methods in Figure A10. Regular decoding [34] often produces object hallucinations, especially in the latter part of the outputs. In contrast, utilizing VL-SAE effectively mitigates the object hallucinations by enhancing the vision-language alignment at the concept level during the decoding process.

### B.5.3 Visualization of the Representations and Learnable Weights

Figure A11 illustrates the t-SNE [58] visualization for the vision-language representations of VLMs and the learnable weights in the encoder of the VL-SAE. We observe that the majority of learnable weights are positioned midway between vision and language representations, rather than being skewed toward either modality. This position enables the corresponding neurons to exhibit consistent activation values across different modalities, which helps map the vision-language representations into a unified concept set.

Table A16: Ablation studies on the hyperparameter $\alpha_l$ of Equation 11. All experiments are conducted on POPE benchmark with LLaVA 1.5.

| Setting | 0.5 | 0.6 | 0.7 | 0.8 | 0.9 |
|---|---|---|---|---|---|
| Random | 85.29 | 85.32 | **85.50** | 85.41 | 85.31 |
| Popular | 84.09 | 84.15 | **84.37** | 84.23 | 84.12 |
| Adversarial | 81.92 | 82.10 | **82.29** | 81.95 | 81.93 |

Table A17: Ablation studies on the impact of the CC3M dataset.

| Method | Base Model | Base Model+Fine-tuning | Base Model+VL-SAE |
|---|---|---|---|
| Mean Accuracy | 69.8 | 69.7 | 70.4 |

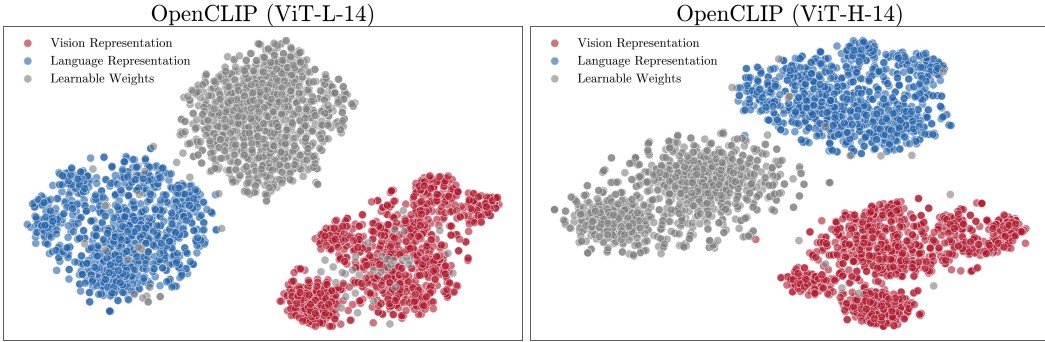

Figure A11: Visualizations of the vision-language representations in the VLMs and the learnable weights in the VL-SAE encoder. Experiments are conducted with t-SNE [58] for dimension reduction.

# C  Discussion

## C.1  Criterion of Representation Selection in LVLMs

We extract the representations from the outputs of the 29-th layer in LLaVA 1.5 and the 26-th layer in Qwen-VL. These layers are chosen because the representations in the top layers tend to be more interpretable in terms of concepts [23]. Furthermore, at these layers, both vision and language representations preserve their respective information effectively. As shown in Figure A12, starting from the 27-th layer of Qwen-VL, the distributions of the two types of representations begin to converge, leading to a potential loss of distinct visual information. This convergence hinders the process of leveraging vision representations to constrain language representations for reducing hallucinations.

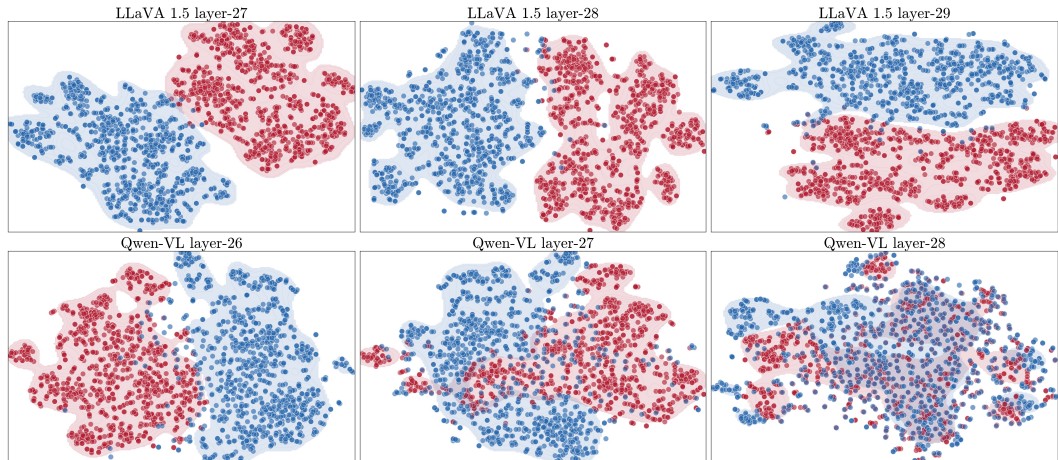

Figure A12: Visualizations of the vision-language representations in different layers of VLMs. Experiments are conducted with t-SNE [58] for dimension reduction.

## C.2 Metrics for Evaluating the Concept Set

For the quantitative evaluation of concept collections, we utilized CLIP similarity for automated assessment in Figure 3 and Table 3. However, the numerical differences in CLIP similarity might not entirely capture the improvements in concept quality achieved by VL-SAE relative to other methods. This is because the numerical gap in CLIP similarity between semantically similar and dissimilar image-text pairs can sometimes be minimal. Therefore, it is important to establish a more comprehensive evaluation mechanism for SAEs based on multi-modal representations in the future.

## C.3 Discussions with attribution-based methods

We discuss and compare VL-SAE, a concept-based approach, with attribution-based methods [67, 4] for VLMs, enabling users to select the most appropriate method based on specific application scenarios. Attribution-based methods generate a heatmap for each input sample to indicate the importance of individual components of the input data (*e.g.*, image pixels), whereas VL-SAE constructs a concept set across varying samples to interpret the semantics of their hidden representations.

Based on their paradigms, we compare them in the perspective of *intuitiveness*, *expressivity* and *ability to improve model performance*. *(i)* Intuitiveness. Compared with VL-SAE, attribution-based methods offer a more intuitive interpretation, as they visualize attribution scores directly on input images. *(ii)* Expressivity. Compared with attribution-based methods, VL-SAE exhibits stronger expressivity as it reveals how the model understands the semantics of input data with concepts, rather than simply providing numerical information to identify the important image regions. *(iii)* Ability to improve model performance. In contrast to attribution-based methods that only serve as an observer for model prediction, VL-SAE can act as an intervener to improve model performance by modifying hidden representations during inference.

## C.4 Limitation Discussion

Despite successfully correlating the semantics of vision-language representations with a unified concept set, the VL-SAE encounters the same challenges as the SAE architectures for other models (*e.g.*, LLMs [56] and vision models [12, 5]), such as dead neurons [15], limited relational modeling, and the impact of high-frequency neurons on interpretations as shown in Figure A8. Additionally, the strategy of enhancing the alignment mechanism with VL-SAE is simple and fails to fully unlock the potential of the unified concept set. To address these limitations, we anticipate future work to design SAEs that encourage the neurons to be uniformly activated, incorporate relational modeling [53, 54, 52] among concepts, and enable more effective integration into the prediction processes of VLMs.

