# OpenReview forum: "VL-SAE: Interpreting and Enhancing Vision-Language Alignment with a Unified Concept Set"
_NeurIPS.cc/2025/Conference — NeurIPS 2025 poster_

### Official Review · Reviewer_obSK · 2025-06-18

**Clarity:** 4
**Significance:** 3
**Originality:** 3
**Rating:** 4
**Confidence:** 4

**Summary:**

This paper aims to solve a problem that current methods cannot map the semantics of vision and language representations into a unified concept set, when interpreting representations of VLMs by mapping their semantics to concepts. The paper proposed to apply a shared SAE (sparse autoencoder) for both modalities. Besides the interpretation method evaluation, the author also provides two applications for enhancing the alignments of both the CVLMs and LVLMs.

**Questions:**

1. In the introduction, it said, “This limited comprehension hinders our ability to analyze and address the misalignment cases, such as hallucinations. To tackle this challenge, existing methods interpret representations of VLMs by mapping their semantics to concepts.”, There are other interpretation methods that explain the matching of image-text pair by attention heat maps, such as Grad-ECLIP[1], and GAE[2]. Should discuss the Pros and Cons of explanation via concepts and attention maps.

[1] Zhao C, Wang K, Zeng X, et al. Gradient-based visual explanation for transformer-based clip[C]//International Conference on Machine Learning. PMLR, 2024: 61072-61091.

[2] Chefer H, Gur S, Wolf L. Generic attention-model explainability for interpreting bi-modal and encoder-decoder transformers[C]//Proceedings of the IEEE/CVF international conference on computer vision. 2021: 397-406.

2. My main concern about the interpretation methods using SAE is that this kind of method seems to build a way to find similar samples for the given image-text pair in the dataset. We may find some regular patterns from the found set, but this kind of method still does not provide which concepts/objects in the specific image are matched with which words in the text description, which, in my opinion, is a more meaningful interpretation way.
3. Will the number of concepts be limited by the number of hidden neuron activations? Based on A.1, 100 concepts are randomly selected to conduct the evaluation. Then, how many concepts will be learned in total?
4. In Figures 5 and 6, it’s unclear about the relationship between the image examples and the text under the image. For example, in Fig. 5, in the concept of image, there are three images, with the following text: “Motorcycle and scooter next to each other.” Are the three images and the text selected from the same node? So why is there only one text while three images? Moreover, there seems to be no scooter in these images, why?
5. In the enhancing vision-language alignment of CVLMs part, I didn’t find the training details, like the training datasets, and not sure whether the training set will influence the zero-shot performance.

I will consider increasing the rating after the questions are solved.

**Ethical Concerns:**

["NO or VERY MINOR ethics concerns only"]

**Final Justification:**

Thank you for the rebuttal. This resolves majority of my concerns. I will keep my score.

**Limitations:**

Yes.

**Quality:**

3

**Strengths And Weaknesses:**

Strengths:
- The paper is well-written and easy to follow.
- The methodology is displayed clearly and thoroughly.
- Comprehensive experiments are conducted.

Weeknesses:
see the questions.

---

> ### Author Rebuttal · Authors · 2025-07-31
>
> We would like to thank you for the insightful comments! We will diligently follow your guidance to enhance the clarity and comprehensibility of our discussions.
>
> ## Q1: Discussions with attribution-based methods.
>
> Thank you for sharing these valuable papers, which significantly enrich our discussion and enable a more comprehensive analysis!
>
> Following your suggestions, we first provide the essential paradigm of attribution-based and concept-based methods. Then we compare them in terms of intuitiveness, expressivity, and ability to improve model performance.
>
> 1. **Paradigm**.
>
>     - Attribution-based methods generate a heat map for each sample to represent the importance of input data (e.g., image pixels).
>
>     - Concept-based methods construct a concept set for varying samples to interpret the semantics of their hidden representations.
>
> 2. **Comparisons**.
>
>     - **Intuitiveness**. Compared with concept-based methods, `attribution-based methods offer a more intuitive interpretation`, as they visualize attribution scores directly on input images.
>
>     - **Expressivity**. Compared with attribution-based methods, `concept-based methods exhibit stronger expressivity` as they reveal how the model understands the semantics of input data with concepts, rather than simply providing numerical information to identify the important image regions.
>
>     - **Ability to improve model performance**. In contrast to attribution-based methods that only serve as an observer for model prediction, `concept-based methods can act as an intervener to improve model performance` by modifying hidden representations during inference (e.g., our experiments in Sections 4.3).
>
> **Revision**. We will incorporate the above discussion and previous studies (i.e., Grad-ECLIP and GAE) into the revision.
>
> ## Q2: Discussions of the interpretation form.
>
> We appreciate your insightful comments that inspire us to incorporate sample-specific concepts into SAEs for more intuitive interpretations.
>
> To address your concern, we first analyze the Pros and Cons of using sample-specific concepts. Then, we provide potential improvements to integrate their advantages into SAEs.
>
> 1. **Pros**: `Extracting concepts from input samples (i.e., image regions, words) can offer intuitive interpretations` similar to those you described.
> 2. **Cons**: However, the sample-specific property `makes it challenging to analyze and intervene in the predictions across different samples`, which is important in the field of mechanism interpretability [R1].
> 3. **Potential improvements**: Despite SAEs offering the analysis and intervention across samples, we agree with your suggestions that they should be improved to provide more intuitive interpretations. We present this potential improvement as part of future work:
>
>     - Given an input sample, we plan to build sample-specific concepts and correlate them with multi-modal concepts of VL-SAE. With the correlation, the VL-SAE concepts serve as a bridge, enabling the match of image regions and words.
>
> **Revision**: We will incorporate the above discussions in the revision, and attempt to improve the interpretation form of VL-SAE in our future work.
>
> ## Q3: Number of concepts in VL-SAE
>
> Thank you for your careful reading of our paper!
>
> 1. The number of concepts is upper bounded by the number of hidden neurons (denoted by $h$ in L216).
> 2. The number of concepts is influenced by the training process and SAE architectures. This stems from the inevitable existence of dead neurons in SAEs [R2], which are never activated. We provide the number of concepts within different SAEs based on LLaVA 1.5 with 4096 feature dimensions. `VL-SAE learns more concepts compared to existing methods`.
>
>     | Method | Hidden Neuron         | Dead Neuron | Concept Number |
>     | ------ | --------------------- | ----------- | -------------- |
>     | SAE-D  | 32768 (4096$\times$8) | 54          | 32714          |
>     | SAE-S  | 32768 (4096$\times$8) | 46          | 32722          |
>     | VL-SAE | 32768 (4096$\times$8) | **15**      | **32753**      |
>
> 3. For evaluation, we observe that the metrics `showed no significant differences when more than 100 concepts are used`. Since the evaluation needs to be repeated multiple times, we select 100 concepts in each iteration for efficient evaluation.
>
>     | Concept Num.              | 100    | 200    | 300    | 400    | 500    |
>     | ------------------------- | ------ | ------ | ------ | ------ | ------ |
>     | Intra Similarity (SAE-S)  | 0.2200 | 0.2257 | 0.2281 | 0.2245 | 0.2247 |
>     | Intra Similarity (VL-SAE) | **0.2445** | **0.2508** | **0.2489** | **0.2476** | **0.2469** |
>     | Inter Similarity (SAE-S)  | 0.1347 | 0.1293 | 0.1300 | 0.1306 | 0.1308 |
>     | Inter Similarity (VL-SAE) | **0.1230** | **0.1214** | **0.1182** | **0.1181** | **0.1195** |
>
> **Revision**. We will incorporate the above analysis into Section 4 of the revision.
>
> ## Q4: Clarifications of Figure 5 and Figure 6.
> Thank you for your comments that help us refine the illustration of Figures 5 and 6!
>
> 1. `The three images and the text are maximum activation samples of the same hidden neuron`. We represent each concept with samples that maximally activate its corresponding neurons in SAE, following [R3]. Given that VL-SAE can process multi-modal inputs, we choose both images and texts to represent each concept.
> 2. `The number of images and texts is not strictly fixed`. For example, we use two images and two sentences to represent concepts in Figure 4. We present three images and one sentence per concept in Figures 5 and 6 due to page limitations.
> 3. `The concept semantics in SAE are represented by the summarization of its maximum activation samples`. Using a single sentence may introduce bias, like "scooter" as you mentioned. We provide more texts of this concept as follows:
>     - "Motorcycle at meeting for bikers an annual event held"
>     - "Motorcycle parked by the side"
>     - "Motorcycle stood on its rest parked on the pavement"
>     - "Motorcycle at a rally or meeting"
>
>     We can observe that this concept can be summarized as a semantic related to motorcycles.
>
> **Revision**: In contrast to displaying maximum activation texts, we will employ LLMs to summarize these texts following [R4] to provide a clearer visualization in Figures 5 and 6.
>
> ## Q5: Implementation details of Table 2.
> Thank you for your careful reading. We would like to clarify that `VL-SAE can enhance VLMs without fine-tuning on downstream tasks`. The VL-SAE is only pre-trained with the CC3M dataset self-supervisedly.
>
> To address your concern, we investigate the influence of CC3M on downstream tasks by comparing the base model (CLIP-ViT-B/16), the base model fine-tuned by CC3M, and the base model equipped with VL-SAE.
>
> | Method        | Base Model | Base Model+Fine-tuning | Base Model+VL-SAE |
> | ------------- | ---------- | ---------------------- | ----------------- |
> | Mean Accuracy | 69.8       | 69.7                   | **70.4**          |
>
> 1. `Fine-tuning the base model with CC3M cannot improve performance`. This is because CC3M is a subset of the VLM pre-training dataset. It does not provide the VLM with any additional task-relevant information.
>
> 2. The superior performance of VL-SAE indicates that `the improvement stems from the mapping from representations to concepts`, rather than the information of CC3M.
>
> **Revision**. We will incorporate the above experiments and implementation details into Section 4.3 of the revision.
>
> [R1] Open Problems in Mechanistic Interpretability, Arxiv'25.
>
> [R2] Scaling and evaluating sparse autoencoders, ICLR'25.
>
> [R3] Sparse autoencoders reveal selective remapping of visual concepts during adaptation, ICLR'25.
>
> [R4] Sparse autoencoders find highly interpretable features in language models, ICLR'24

---

### Official Review · Reviewer_i9F9 · 2025-06-25

**Clarity:** 4
**Significance:** 3
**Originality:** 4
**Rating:** 5
**Confidence:** 4

**Summary:**

This paper focuses on interpreting the alignment mechanism between vision and language representations of pre-trained VLMs. The authors propose VL-SAE, a novel sparse autoencoder architecture for multi-modal representations, which maps vision-language semantics into a unified concept set. In experiments, the proposed method is evaluated across multiple pre-trained VLMs. Furthermore, two strategies are proposed to enhance model performance on downstream tasks using VL-SAE, including the zero-shot classification for CVLMs and hallucination elimination LVLMs.

**Questions:**

According to the above weaknesses, I recommend that the authors provide further ablation studies focusing on sparsity constraints and the architecture of the auxiliary autoencoder, as well as report the additional costs introduced by VL-SAE.

**Ethical Concerns:**

["NO or VERY MINOR ethics concerns only"]

**Final Justification:**

The authors propose a novel SAE architecture for multi-modal representations, enabling the interpretation of their alignment mechanism.   Extensive experiments have demonstrated the effectiveness of VL-SAE. The rebuttal has addressed all my concerns about this work. So I recommend it for acceptance.

**Limitations:**

Yes

**Quality:**

3

**Strengths And Weaknesses:**

Strengths:
1.	The authors propose a novel SAE architecture for multi-modal representations, enabling the interpretation of their alignment mechanism.
2.	The proposed method can be applied to models with different architectures, such as CVLMs and LVLMs, demonstrating strong scalability.
3.	Beyond interpreting representations, the authors propose two strategies to enhance the alignment of pre-trained VLMs, which lead to improved performance on downstream tasks.
4.	This paper is well-written and easy to follow.
5.	The authors conduct extensive experiments to demonstrate the effectiveness of VL-SAE.
Weaknesses:
1.	As all SAEs are pre-trained on the same dataset, it would be interesting to explore how the performance of VL-SAE varies with different volumes of pre-training data.
2.	While the proposed approach can achieve performance improvements, it is important to consider the additional cost introduced by integrating VL-SAE into the model inference pipeline.
3.	Why does the auxiliary autoencoder described in Section 3.2 also require the use of modality-specific encoders and decoders?
4.	It would be better to present the qualitative comparison between SAE-S, SAE-D and VL-SAE of Figure A8 in the main text.
5.	I am interested in exploring the potential effects of utilizing L1 loss as the sparsity constraint, as opposed to employing Top-k selection.

---

> ### Author Rebuttal · Authors · 2025-07-31
>
> We would like to thank you for the detailed comments! We will diligently follow your guidance to further improve our work and manuscripts.
>
> ## W1: Experiments with different volumes of pre-training data.
> Following your suggestion, we train VL-SAE with different volumes of the dataset.
>
> - **Experiments**: We train VL-SAE models with different proportions of the CC3M dataset and report their Intra-Similarity and Inter-Similarity.
>
>     | Data Percentage                 | 10\%    | 30\%    | 50\%    | 70\%    | 100\%       |
>     | ------------------------------- | ------ | ------ | ------ | ------ | ---------- |
>     | Intra-Similarity ($\uparrow$)   | 0.2029 | 0.2129 | 0.2196 | 0.2222 | **0.2299** |
>     | Inter-Similarity ($\downarrow$) | 0.1597 | 0.1306 | 0.126  | 0.1256 | **0.1220** |
>
>
> - **Analysis**: We observe that VL-SAE demonstrates strong scalability. As the volume of data increases, the Intra-Similarity (0.2029 $\rightarrow$ 0.2299) and Inter-Similarity (0.1597 $\rightarrow$ 0.1220) of the concepts learned by VL-SAE improve accordingly.
>
> **Revision**. We will incorporate the above experiments and analysis into the main text.
>
> ## W2: Additional cost of incorporating VL-SAE.
> Thank you for your insightful comments! Following your suggestion, we provide the additional cost of incorporating VL-SAE in terms of memory usage, computations, and inference speed.
>
> - **Memory usage**:
>
>     | Model          | Total Param. | VL-SAE Param. | Proportion |
>     | -------------- | ------------ | ------------- | ---------- |
>     | LLaVA-1.5      | 7529M        | 467M          | 0.06       |
>     | OpenCLIP-ViT-H | 1011M        | 25M           | 0.02       |
>
> - **Computation**:
>
>     | Model          | Total FLOPs | VL-SAE FLOPs | Proportion |
>     | -------------- | ----------- | ------------ | ---------- |
>     | LLaVA-1.5      | 14.03G      | 0.875G       | 0.06       |
>     | OpenCLIP-ViT-H | 3.15G       | 0.034G       | 0.01      |
>
> - **Inference Speed**: The experiments are provided in Table A6 in the appendix.
>
> - **Analysis**: According to the above experimental results, the additional cost of incorporating VL-SAE is nearly negligible. It is cost-effective to allocate less than 6\% of the parameter and computational resources to achieve interpretability in the model inference process.
>
> **Revision**. We will incorporate the above experiments and analysis into the revision.
>
> ## W3: Clarifications of the auxiliary autoencoder.
>
> Thank you for your valuable suggestions! We will explain the usage of modality-specific encoders and decoders in terms of motivation and performance.
>
> - **Motivation**: First, the auxiliary autoencoder should possess strong expressive capabilities, as it is required to directly reconstruct the high-dimensional vision-language representations of LVLMs. Therefore, we design separate encoders and decoders for each modality, rather than relying on a shared model, to enhance the model's performance on this reconstruction task. Second, inspired by the successful training protocol of CLIP, our approach aims to establish a correlation between the semantic similarity and the representation cosine similarity. Therefore, we adopt a modality-specific architecture similar to that of CLIP.
>
> - **Performance**: To address your concern, we provide comparisons between the shared and modality-specific strategies. We observe that employing a modality-specific strategy for the auxiliary autoencoder results in lower reconstruction loss and improved concept quality of the corresponding VL-SAE.
>
>     | Method            | Recon. Loss | Intra-Similarity ($\uparrow$) | Inter-Similarity ($\downarrow$) |
>     | ----------------- | ----------- | ---------------- | ---------------- |
>     | share             | 0.0047      | 0.2157           | 0.2021           |
>     | modality-specific | **0.0034**      | **0.2283**           | **0.1762**           |
>
> **Revision**. We will incorporate the above analysis into the revision.
>
> ## W4: Moving Figure A8.
> Thank you for your suggestion! We will present Figure A8 in Section 4.2 in the revised paper.
>
> ## W5: Utilizing L1 loss for sparsification.
>
> - **Experiments**: Following your suggestion, we compare the performance of utilizing Top-k and L1 loss for sparsification. We set the coefficient of L1 loss to 1e-4.
>
>     | Method | Intra-Similarity ($\uparrow$) | Inter-Similarity ($\downarrow$) |
>     | ------ | ---------------- | ---------------- |
>     | L1     | 0.2142           | 0.1809           |
>     | Top-K  | **0.2442**           | **0.1373**           |
>
> - **Analysis**: We find that utilizing Top-K for sparsification achieves better performance than L1 loss. This is because the L1 loss cannot directly control the representation sparsity. Our results are consistent with previous studies [R1], which also indicate that Top-K sparsification is a more effective approach.
>
> **Revision**. We will incorporate the above experiments and analysis into the revision.
>
> [R1] Scaling and evaluating sparse autoencoders, ICLR'25.

---

### Official Review · Reviewer_5uDR · 2025-07-03

**Clarity:** 3
**Significance:** 3
**Originality:** 3
**Rating:** 4
**Confidence:** 4

**Summary:**

Vision-Language models (VLMs) achieve remarkable performance on various downstream tasks, which can be attributed to the inductive biases which facilitate vision-language alignment. However, it remains a substantial challenge to interpret vision-language alignment. This work proposes VL-SAE to map the semantics of vision-language representations to a unified concept set and thus interpret the alignment mechanism of VLMs. Further experiments apply VL-SAE to a set of VLMs to show the high quality of concept sets from VL-SAE. Last, this work applies VL-SAE to interpret Contrastive VLMs’ predictions by visualizing the activated concepts, and also to further improve model’s performance on zero-shot image classification and hallucination benchmarks.

**Questions:**

1. L209, what is $\theta$?
2. L254-255, the authors mention that the training of autoencoders is skipped. Does it mean the SAEs are trained directly on the fly? Also, does training of the autoencoder hurt CVLMs performance?

**Ethical Concerns:**

["NO or VERY MINOR ethics concerns only"]

**Final Justification:**

The rebuttal resolves majority of my concerns, especially the clarifications of the auxiliary autoencoders and the additional experiments on CC3M and human study. I hope the authors can incorporate my comments into the final version.

**Limitations:**

yes

**Paper Formatting Concerns:**

None.

**Quality:**

3

**Strengths And Weaknesses:**

**Strengths**
1. The idea of having a unified concept set to explain vision-language alignment is interesting and important.
2. Experimental results show the effectiveness of VL-SAE on learning unified and diverse concept sets.
3. Experimental results show that VL-SAE can further improve VLM’s performance on zero-shot image classification and hallucination benchmark.
4. Qualitative results further demonstrate the unified concepts’ quality.

**Weaknesses**
1. L209, the paper mentions triangle inequality for several times, and claims that the distance g satisfies the triangle inequality. However, there is no further clarification or proof to support this.
2. L289-290, what is the difference between “-Auxiliary autoencoder” and the baseline (Table 1)? If I understand correctly, “-Auxiliary autoencoder” means no training of autoencoders, which reflects the setting in L254-255 where CVLMs skip the training of autoencoders as well. Should we consider adding this training phase in?
3. For Figure 5, it is great to show qualitative results of the activated concepts. However, it lacks human studies to quantitatively measure the benefits of using VL-SAEs to interpret vision-language alignment.
4. L312, does $\alpha$ have to be task-specific? Is that possible to generalize this across tasks? How is this value determined for each task?
5. For Table 2, it is unclear whether it is the CC3M training that brings the performance improvement. For the non VL-SAE rows, do these models also further finetuned on CC3M dataset? Ideally, there should be 3 rows per bracket: original model, original model + CC3M finetuning, original model + CC3M finetuning with VL-SAE.
6. For Equation 11, $\alpha$ overlaps with that in Equation 10. How is this $\alpha$ value determined? How will it change model’s performance?
7. L335 (Table 3), similar to the above, is VCD also further finetuned on CC3M dataset?

---

> ### Author Rebuttal · Authors · 2025-07-31
>
> We would like to thank you for the constructive comments! We will diligently follow your suggestions to further improve our manuscripts.
>
> ## W1: Proof of L209.
> Thank you for your careful reading! As illustrated in line 207, `the function g is defined as the Euclidean distance between normalized vectors, which inherently satisfies the triangle inequality [R1]`. To address your concerns, we provide the proof as follows.
>  - **Proof**: Given $\mathbf{a},\mathbf{b},\mathbf{c}\in\mathbb{R}^d$, Let's denote their normalized vectors as $\bar{\mathbf{a}}=\frac{\mathbf{a}}{\Vert\mathbf{a}\Vert_2}, \bar{\mathbf{b}}=\frac{\mathbf{b}}{\Vert\mathbf{b}\Vert_2}, \bar{\mathbf{c}}=\frac{\mathbf{c}}{\Vert\mathbf{c}\Vert_2}$. The function $g(\mathbf{a},\mathbf{b})$ can be represented as the L2-norm of $\mathbf{x}=\bar{\mathbf{a}}-\bar{\mathbf{b}}$,
>  $$g(\mathbf{a},\mathbf{b})=\Vert\frac{\mathbf{a}}{\Vert\mathbf{a}\Vert_2}-\frac{\mathbf{b}}{\Vert\mathbf{b}\Vert_2}\Vert_2 =\Vert\bar{\mathbf{a}}-\bar{\mathbf{b}}\Vert_2=\Vert\mathbf{x}\Vert_2$$
>  Similarly, we can represent $g(\mathbf{b},\mathbf{c})$ with $\mathbf{y}=\bar{\mathbf{c}}-\bar{\mathbf{b}}$,
>  $$g(\mathbf{b},\mathbf{c})=\Vert\frac{\mathbf{c}}{\Vert\mathbf{c}\Vert_2}-\frac{\mathbf{b}}{\Vert\mathbf{b}\Vert_2}\Vert_2 =\Vert\bar{\mathbf{c}}-\bar{\mathbf{b}}\Vert_2=\Vert\mathbf{y}\Vert_2$$
>  $g(\mathbf{b},\mathbf{c})$ with $\mathbf{y}=\bar{\mathbf{c}}-\bar{\mathbf{b}}$,
>  $g(\mathbf{a},\mathbf{c})$ can be represented by the Euclidean distance between $\mathbf{x}$ and $\mathbf{y}$,
>  $$g(\mathbf{a},\mathbf{c}) =\Vert\bar{\mathbf{a}}-\bar{\mathbf{c}}\Vert_2=\Vert(\bar{\mathbf{a}}-\bar{\mathbf{b}})-(\bar{\mathbf{c}}-\bar{\mathbf{b}})\Vert_2=\Vert\mathbf{x}-\mathbf{y}\Vert_2$$
>  According to the Cauchy-Schwarz inequality, we get:
>  $$\vert\Vert\mathbf{x}\Vert_2-\Vert\mathbf{y}\Vert_2\vert\le \Vert\mathbf{x}-\mathbf{y}\Vert_2$$
>  This inequality can derive Equation (5) in our paper.
>  $$\vert g(\mathbf{a},\mathbf{b})-g(\mathbf{b},\mathbf{c})\vert\le g(\mathbf{a},\mathbf{c})$$
>
> **Revision**. We will integrate the aforementioned proof into the Appendix.
>
> ## W2 \& Q2: Clarification of the auxiliary autoencoder.
> Thank you for your comments that help us better clarify our implementation details! To avoid misunderstanding, we first introduce the components of our framework and then clarify the content of L289–290 and L254–255.
>
> 1. **Components of our framework**. There are `two autoencoders` in our framework. One is the auxiliary autoencoder, designed for LVLMs to transform their implicitly aligned multi-modal representations into explicitly aligned representations that correlate semantic similarity with cosine similarity. The other is the VL-SAE with distance-based encoder and modality-specific decoder, designed to interpret explicitly aligned representations.
> 2. **Clarification of L289-290**. The "-Auxiliary autoencoder" means directly training the VL-SAE with implicitly aligned representations of LVLMs rather than explicitly aligned representations of the auxiliary autoencoder. `Compared with the baseline, the "-Auxiliary autoencoder" version replaces the standard SAE with the VL-SAE`.
> 3. **Clarification of L254-255**. We inadvertently omitted writing the "auxiliary" in L254-255.
> In CVLM, training an auxiliary autoencoder is unnecessary because the CVLM representations are already explicitly aligned. `The VL-SAE for CVLMs still requires training`.
>
> **Revision**. We will revise "training of the autoencoder" to "training of the auxiliary autoencoder" in L254–255, and clarify the role of the auxiliary autoencoder in Section 3.2.
>
> ## W3: Human evaluations.
> We appreciate your suggestions that help us improve the comprehensiveness of our evaluations.
>
> - **Experiments**. We randomly select 100 concepts from SAE-D, SAE-S, and VL-SAE, respectively, and invite 10 participants to evaluate these concepts. For each iteration, participants are required to assess the quality of three given concepts and select the one with the highest quality. The method corresponding to the selected concept is awarded one point. The concept quality of these methods is measured by the average score assigned by all participants.
>
>     | Method | OpenCLIP-ViT/H | LLaVA 1.5 |
>     | ------ | -------------- | --------- |
>     | SAE-D  | 0.6            | 0.1       |
>     | SAE-S  | 33.5           | 35.8      |
>     | VL-SAE | **65.9**       | **64.1**  |
>
> - **Analysis**. `The concepts learned by VL-SAE possess higher quality than those learned by other methods`. This comparison further highlights the effectiveness of VL-SAE.
>
> **Revision**. The scale of current human evaluation is constrained by the limited rebuttal period. Based on the above experiments, we plan to expand the evaluation scale and incorporate it into the revision.
>
> ## W4: Selection of hyper parameters $\alpha$.
> This question is valuable and prompts us to enhance the ability of VL-SAE in improving performance.
>
> We first provide accuracy comparisons under three settings: baseline ($\alpha=0$), task-agnostic ($\alpha=0.1$), and task-specific ($\alpha$ determined by tasks). Then, we address your concerns by analyzing the results.
>
> | Base Model    | OpenCLIP-ViT-L | OpenCLIP-ViT-H |
> | ------------- | -------------- | -------------- |
> | baseline      | 72.2           | 76.9           |
> | task-agnostic | 72.4           | 77.2           |
> | task-specific | **72.9**       | **77.8**       |
>
> 1. The task-agnostic scheme performs better than the baseline (72.4\% v.s. 72.2\% on ViT-L, and 77.2\% v.s. 76.9\% on ViT-H). This indicates that `$\alpha$ does not need to be task-specific for performance improvement and can generalize across tasks`.
> 1. However, these tasks originate from different domains and exhibit distinct preferences for hyperparameters [R2], as evidenced by the best performance of the task-specific scheme. `If the objective is to achieve optimal performance, the $\alpha$ values of each task should be determined individually`,
> 3. To determine the $\alpha$ value of each task, we conduct experiments with multiple values $\alpha\in[0,1]$ at intervals of 0.1 and select the one that performs the best. Note that `the cost of conducting multiple experiments is very low`. For instance, when using the Food-101 dataset, each experiment on a single NVIDIA 4090 GPU takes only approximately 90 seconds.
>
> **Revision**. We will incorporate the above experiments into the revision and improve the performance of the task-agnostic scheme in future work.
>
> ## W5: Effects of CC3M training.
> Thank you for your detailed feedback! `The CC3M dataset is employed to integrate VL-SAE as a general plug-in for VLMs, rather than to augment the intrinsic capabilities of these VLMs`. In our training strategies, the VLM parameters remain frozen without any adjustments.
>
> To address your concern, we provide comparisons between the base model (CLIP-ViT-B/16), the base model fine-tuned by CC3M, and the base model equipped with VL-SAE.
>
> | Method        | Base Model | Base Model+Fine-tuning | Base Model+VL-SAE |
> | ------------- | ---------- | ---------------------- | ----------------- |
> | Mean Accuracy | 69.8       | 69.7                   | **70.4**          |
>
> 1. `Fine-tuning does not lead to improved performance on downstream tasks`. This is because CC3M serves as a subset of the VLM pre-training dataset. It does not provide the base model with any additional task-relevant information.
> 2. The superior performance of VL-SAE compared to the fine-tuning model indicates that `the improvement stems from the mapping from representations to concepts`, rather than the incorporation of CC3M.
>
> **Revision**. We will incorporate the above experiments and analysis into the revision.
>
> ## W6: Explanations of the hyperparameters in Equation 11.
> Thank you for pointing out the confused notations.
>
> - **Notation Overlap**. We will re-denote the $\alpha$ in Equation 11 as $\alpha_l$, and denote the hyperparameter in Equation 10 as $\alpha_c$ to eliminate the notation overlap.
> - **Value Determination**. We conduct experiments by selecting values at intervals of 0.1 within the range of $[0.5-0.9]$, and the one with the best performance is selected.
> - **Performances of varying $\alpha_l$ values**. We report the performance under different $\alpha_l$ values. The highest performance across different settings is achieved at $\alpha_l=0.7$.
>
>     | Setting     | 0.5   | 0.6   | 0.7   | 0.8   | 0.9   |
>     | ----------- | ----- | ----- | ----- | ----- | ----- |
>     | Random      | 85.29 | 85.32 | **85.50** | 85.41 | 85.31 |
>     | Popular     | 84.09 | 84.15 | **84.37** | 84.23 | 84.12 |
>     | Adversarial | 81.92 | 82.10 | **82.29** | 81.95 | 81.93 |
>
> **Revision**. We will employ distinct notations to represent the hyperparameters of Equations 10 and 11, and integrate the above experiments into the revision.
>
> ## W7: Clarification of Table 3.
> Thank you for your comments! As a training-free method without learnable parameters, VCD cannot be fine-tuned on CC3M.
>
> However, as a subset of the pre-training data for VLMs, CC3M is used to endow VL-SAE with the ability to process VLM representations, which is irrelevant to downstream tasks. Since `neither VL-SAE nor VCD has been exposed to information relevant to downstream tasks`, the comparison in Table 3 is fair.
>
> **Revision**. We will further clarify the usage of the CC3M dataset in the revision.
>
> ## Q1: Explanations of L209.
> Thank you for your careful reading! $\theta_{\mathbf{x},\mathbf{y}}$ denotes the angle between two vectors $\mathbf{x}$ and $\mathbf{y}$, and we use $cos(\theta_{\mathbf{x},\mathbf{y}})$ to represent the cosine similarity between these two vectors.
>
> **Revision**. We will clarify the meaning of this notation in Section 3.3 of the revision.
>
> [R1] The Cauchy-Schwarz Inequality: A Geometric Proof. The American Mathematical Monthly, 1989.
>
> [R2] Scaling \& shifting your features: A new baseline for efficient model tuning, NeurIPS'22.

---

> > ### Comment · Reviewer_5uDR · 2025-08-05
> >
> > Thank you for the rebuttal. This resolves majority of my concerns and I hope the authors can incorporate my comments and other reviewers' into the camera ready version. I will keep my score.

---

> > > ### Author Response · Authors · 2025-08-06
> > >
> > > Dear Reviewer 5uDR,
> > >
> > > Thank you very much for your positive response and continued support of our work. We are glad that the additional clarifications and experiments addressed your concerns. We will make sure to incorporate the comments of all reviewers into the revised version of this paper.
> > >
> > > Once again, thank you for your valuable feedback and constructive suggestions.
> > >
> > > Sincerely,
> > > Authors of Submission 1904

---

### Official Review · Reviewer_LRaD · 2025-07-07

**Clarity:** 2
**Significance:** 2
**Originality:** 2
**Rating:** 3
**Confidence:** 3

**Summary:**

The authors propose Explicit Representation Alignment and VL-SAE to enhance interpretation of VLLM and address concept mismatch in Large Vision-Language Models. Each neuron in the hidden layer correlates to a concept represented by semantically similar images and texts, thereby interpreting these representations with a unified concept set. To establish the neuron-concept correlation, they encourage semantically similar representations to exhibit consistent neuron activations during self-supervised training. Experiments across multiple VLMs (e.g., CLIP, LLaVA) demonstrate the superior capability of VL-SAE in interpreting and enhancing the vision-language alignment.

**Questions:**

1. I cannot tell the difference in figure 5.
2. Please label the better model for individual class in table 2
3. The workflow of how to get figure 5 and 6 is not clear. Which is input and which is output and why do you put all of these materials here?
4. What is the computation and data scale for training an extra autoencode.
5. Will this work for larger VLMs? Will the data and computation, especially data required scaled accordingly. If so, will the cost be affordable?
6. Some error analysis may be more helpful in the appendix to point future research direction.

**Ethical Concerns:**

["NO or VERY MINOR ethics concerns only"]

**Limitations:**

1. The presentation is hard for me to understand.
2. The author should give a contact workflow of how to utilize this method and align the workflow with experiments like figure 5 and 6.
3. More training details of reproducing this work should be mentioned.

**Paper Formatting Concerns:**

No formatting concern.

**Quality:**

2

**Strengths And Weaknesses:**

Strengths:
1. It proposes a method to explain the correlation between text and vision and avoid concept mismatch.
2. It introduces a distance-based encoder that ensures activation consistency for semantically similar representations via cosine similarity and Euclidean distance constraints. It also uses modality-specific decoders to handle distributional differences between vision and language representations, mitigating concept mismatch.
3. It builds on strong theoretical model.
4. Figure 3 is clear and shows the advantage.

Weakness:
1. The presentation of the paper is not so clear, especially the workflow of the experiments.
2. I cannot get the information that wants to expressed in figure 5 and 6.
3. There is no human evaluation which will reflect real preference in usage.

---

> ### Author Rebuttal · Authors · 2025-07-31
>
> We sincerely thank you for your valuable feedback on our manuscript. Your comments have been instrumental in guiding our future work.
>
> ## W1 \& L1: Explanations of the experiment workflow.
> Thank you for your sincere comments! However, we may not fully understand your specific concerns. To thoroughly address them, we will describe the overall framework of our experiments.
>
> 1. First, we train the VL-SAE for pre-trained VLMs in a self-supervised manner based on their representations (in Section 4.2).
> 2. Then, we integrate VL-SAE into the inference process to interpret the semantics of model representations (in L295-305 for CVLMs, L317-326 for LVLMs).
> 3. Based on the interpretation results, we design methods to intervene in the model predictions via VL-SAE, thereby achieving performance improvement (in L306-316 for CVLMs, L327-337 for LVLMs).
>
> **Revision**. We will emphasize the overall framework of the experiments at the beginning of Section 4 in the revision. Please let us know if your concerns are from other points. We are standing by to address them.
>
> ## W2 \& Q1 \& Q3 \& L2: Explanations of Figure 5 and 6.
> Thank you for your detailed feedback! We will re-introduce Figures 5 and 6 to you from four aspects: the objective, the workflow (Q3, L2), the differences of concepts from different columns (Q1), and the insights we can derive from these figures (W2).
>
> - **Objective**. We present these figures to show the ability of VL-SAE to interpret the alignment of vision-language representations during the prediction process.
>
> - **The workflow in Figures 5 and 6 (Q3, L2)**.
>     1. Given a sample, we feed it into VLMs for inference.
>     2. During the inference process, we extract the vision-language representations from the hidden layers.
>     3. These representations are mapped to human-comprehensible concepts by VL-SAE.
>     4. We display several concepts most relevant to the representation semantics in Figures 5 and 6. Each concept is represented by several images and texts.
>     5. With these concepts correlated to vision and language representations, we can analyze the internal mechanism of the model's prediction for this sample.
>
> - **Differences of concepts from different columns in Figure 5 (Q1)**. The first and second columns display the concepts relevant to vision and text representations, respectively, while the third column shows concepts highly relevant to both modalities.
>
> - **Insights of Figures 5 and 6 (W2)**. In Figures 5 and 6, we `observe several concepts where the two modalities are not aligned`. For example, the "car" concept is irrelevant to the input text in Figure 5, and the "microwave" concept correlates with hallucinated text rather than the input image in Figure 6. These misalignment concepts may affect the multi-modal reasoning during inference. To address this issue, we introduce the methods detailed in L306–316 and L327–337, which effectively mitigate misalignment and lead to improved model performance.
>
> **Revision**. We will describe the workflow of Figures 5 and 6 more clearly, and highlight the misalignment concepts in the figures with a special color to avoid confusion.
>
> ## W3: Human Evaluation Results.
> We appreciate your valuable suggestions that motivate us to provide more comprehensive evaluations.
>
> - **Experiments**. Following your suggestions, we randomly select 100 concepts from SAE-D, SAE-S, and VL-SAE, respectively, and invite 10 participants to evaluate these concepts. For each iteration, participants are required to evaluate the quality of three given concepts and select the one with the highest quality. The method corresponding to the selected concept is awarded one point. The concept quality of these methods is measured by the average score assigned by all participants.
>
>     | Method | OpenCLIP-ViT/H | LLaVA 1.5 |
>     | ------ | -------------- | --------- |
>     | SAE-D  | 0.6            | 0.1       |
>     | SAE-S  | 33.5           | 35.8      |
>     | VL-SAE | **65.9**       | **64.1**  |
>
> - **Analysis**. `The concepts learned by VL-SAE possess higher quality than those learned by other methods`. This comparison further highlights the effectiveness of our proposed VL-SAE.
>
> **Revision**. The scale of current human evaluation is constrained by the limited rebuttal period. Based on the above experiments, we plan to expand the evaluation scale and incorporate it into the revision.
>
> ## Q2: Refining Table 2.
> Thank you for your advice! Following your suggestion, we will label the better model for the individual task in the revision.
>
> ## Q4: Computation and dataset requirements.
> **Computation**: To demonstrate the training efficiency of VL-SAE, we compare it with LoRA, a widely recognized computationally efficient method. The FLOPs of training VL-SAE with OpenCLIP-ViT-B/16 representations are significantly lower than those of LoRA fine-tuning `(0.03G v.s. 1.91G)`. The minimal training cost stems from the small number of parameters in VL-SAE, which is equivalent to approximately two linear layers.
>
> **Dataset Scale**: As illustrated in Lines 247-249, all autoencoders are trained on the CC3M dataset, which comprises `3 million image-text pairs`.
>
> **Usability**: In practice, with the representations of the pre-trained model, we can complete the training of VL-SAE in `less than 10 minutes using a single NVIDIA RTX 4090`, which is efficient and easy to implement.
>
> **Revision**. We will incorporate the comparisons of training FLOPs in the revision.
>
> ## Q5: Scalability and training cost of VL-SAE.
> Thank you for your insightful questions that help us further clarify the low cost and scalability of VL-SAE.
>
> - **VL-SAE works for larger VLMs**. We provide the performance gains achieved by applying VL-SAE to CVLMs of different sizes with the same dataset for pre-training (CC3M). The mean accuracy across various zero-shot image classification tasks in Table 2 is reported as follows.
>
>     | Base Model        | Params. | Original Acc. | +VL-SAE | $\Delta$ Acc. |
>     | ----------------- | ------- | ------------- | ------- | ---------- |
>     | OpenCLIP-ViT-B/32 | 152M    | 68.7          | 69.5    | 0.8        |
>     | OpenCLIP-ViT-B/16 | 177M    | 69.8          | 70.4    | 0.6        |
>     | OpenCLIP-ViT-L/14 | 428M    | 72.2          | 72.9    | 0.7        |
>     | OpenCLIP-ViT-H/14 | 1367M   | 76.9          | 77.8    | 0.9        |
>
>     With the same amount of data, VL-SAE can provide consistent performance gains for models of different sizes. Therefore, the `data scale does not need to be proportional to the model size`.
>
> - **The training cost of VL-SAE is affordable**. We provide the training FLOPs of VL-SAE with models of different sizes, as well as the total training time required on a single NVIDIA RTX 4090.
>
>     | Base  Model   | OpenCLIP-ViT-B/16 | OpenCLIP-ViT-L/14 | OpenCLIP-ViT-H/14 |
>     | ------------- | ----------------- | ----------------- | ----------------- |
>     | FLOPs         | 0.03G             | 0.06G             | 0.10G             |
>     | Training Time | 132s              | 228s              | 446s              |
>
>     The training computations and time of VL-SAE increase with the model size. Nevertheless, benefiting from the small number of parameters, the `pre-training of VL-SAE can be completed in less than 450 seconds using a single NVIDIA RTX 4090`, demonstrating its high computational efficiency.
>
> **Revision**. We will incorporate the above experiments and analysis into the revision.
> ## Q6: Error Analysis.
>
> Thank you for your comments, which have helped us gain perspectives on future research directions through the analysis of failure cases.
>
> - **Current limitations**. As discussed in the limitation section, `although this is not the central focus of our study, VL-SAE encounters the issue of frequently activated neurons, which is prevalent among existing SAEs`. These neurons can be activated by semantically irrelevant samples, reducing the quality of interpretations for these samples.
>
> - **Our Solution**. We have mitigated this issue within the inference process by re-weighting the concept relevancy with the method in Section A.2. The concepts relevant to a kitchen image with and without re-weighting are presented as follows. Due to the inability to upload images, they can only be shown in text form here. The concept corresponding to the high-frequency neuron is denoted by (*), which exhibits irrelevant semantics with the input kitchen image.
>
>     | Before re-weighting                        | After re-weighting                                |
>     | --------------------------------------- | ---------------------------------------------- |
>     | toast popping out of a toaster          | toast popping out of a toaster                 |
>     | hard rock artist performs (*)           | burning on a gas stove in the kitchen          |
>     | burning on a gas stove in the kitchen   | cans in a refrigerator in a restaurant         |
>     | cans in a refrigerator in a restaurant  | this is a gas stove with electric ovens        |
>     | this is a gas stove with electric ovens | gas fueled rings on a domestic cooker |
>
> - **Future Directions**. Although we have mitigated the issues caused by frequently activated neurons at the inference stage, these neurons essentially stem from the training process of SAE. We will address this issue more comprehensively at the training stage in our future work.
>
> **Revision**. We will incorporate the above analysis of failure cases in the appendix.
>
> ## L3: More training details.
> Thank you for your feedback. The details of training VL-SAE are presented in Section 4.1 and A.1. The details regarding the application of VL-SAE to interpret and enhance vision-language alignment are provided in Section A.2~A.4.
>
> Please let us know if the information provided in these sections does not address your concerns. We are standing by to address them.

---

> > ### Author Response · Authors · 2025-08-06
> > **Follow-up on Rebuttal Discussion**
> >
> > Dear Reviewer LRaD,
> >
> > Thank you again for your valuable time and thoughtful feedback on our submission.
> >
> > We would like to kindly follow up to check whether our response has addressed your concerns. If there are any remaining questions or suggestions, we would greatly appreciate the opportunity to clarify further.
> >
> > Your feedback is valuable to us and has been helpful in improving the quality of our work. Thank you again for your engagement.
> >
> > Best regards,
> >
> > The Authors of Submission 1904

---

### Decision · Program_Chairs · 2025-09-17

**Decision:**

Accept (poster)

**Comment:**

This submissions proposes to map the semantics of vision-language representations to a unified concept set and thus interpret the alignment mechanism of VLMs, spanning a bridge between contrastive VLMs like CLIP, which learn explicit alignment, and LVLMs, which do this alignment implicitely. During the review process the paper received mixed reviews with ratings from 3/BR to 5/A. The reviewers appreciated the core idea and the interpretability aspects.

Several weaknesses were raised:
- Presentation and writing, in particular the experimental section which lacks details
- Qualitative results are difficult to evaluate.
- Questions on the origins of gains
- Sensitiviy on the amount of training data
- Comparisons to competing work

Post author/reviewer discussion also raised the problem of the introduction of a new distance function without any ablation or justification of this choice.

The authors provided a rebuttal and attempted answers to some of the concerns, in particular some limited experiments with human evaluation (10 people), sensitivity experiments, which most reviewers found convincing. While the paper still has flaws, the AC judges that the advantages outweight the issues and recommends acceptance.

The authors are requested to provide the additional ablatation in the camera ready version.